# Translocator protein is a marker of activated microglia in rodent models but not human neurodegenerative diseases

Erik Nutma [1,2,22], Nurun Fancy [3,4,22], Maria Weinert [3,22], Stergios Tsartsalis [3,4,5,22], Manuel C. Marzin[1], Robert C. J. Muirhead[3,4], Irene Falk[6,7], Marjolein Breur[1], Joy de Bruin[1], David Hollaus[1], Robin Pieterman[1], Jasper Anink[8], David Story[9], Siddharthan Chandran[9], Jiabin Tang[3,4], Maria C. Trolese [10], Takashi Saito [11], Takaomi C. Saido[12], Katharine H. Wiltshire[3], Paula Beltran-Lobo[13], Alexandra Phillips[3,4], Jack Antel [14], Luke Healy [14], Marie-France Dorion[15], Dylan A. Galloway[15], Rochelle Y. Benoit[15], Quentin Amossé [5], Kelly Ceyzériat[5], Aurélien M. Badina [5], Enikö Kövari[5], Caterina Bendotti [10], Eleonora Aronica [8], Carola I. Radulescu[3,4], Jia Hui Wong [16], Anna M. Barron [16], Amy M. Smith[4,17], Samuel J. Barnes [3,4], David W. Hampton[9], Paul van der Valk[1], Steven Jacobson[6], Owain W. Howell[18], David Baker[19], Markus Kipp [20], Hannes Kaddatz [20], Benjamin B. Tournier [5], Philippe Millet [5,21], Paul M. Matthews [3,4,23], Craig S. Moore [15,23], Sandra Amor[1,19,20,23] ✉ & David R. Owen [3,4,23] ✉

Microglial activation plays central roles in neuroinflammatory and neurodegenerative diseases. Positron emission tomography (PET) targeting 18 kDa Translocator Protein (TSPO) is widely used for localising inflammation in vivo, but its quantitative interpretation remains uncertain. We show that TSPO expression increases in activated microglia in mouse brain disease models but does not change in a non-human primate disease model or in common neurodegenerative and neuroinflammatory human diseases. We describe genetic divergence in the TSPO gene promoter, consistent with the hypothesis that the increase in TSPO expression in activated myeloid cells depends on the transcription factor AP1 and is unique to a subset of rodent species within the *Muroidea* superfamily. Finally, we identify LCP2 and TFEC as potential markers of microglial activation in humans. These data emphasise that TSPO expression in human myeloid cells is related to different phenomena than in mice, and that TSPO-PET signals in humans reflect the density of inflammatory cells rather than activation state.

Neuronal-microglial signalling limits microglial inflammatory responses under homeostatic conditions[1]. The loss of this cross talk in central nervous system (CNS) pathology partly explains why microglia adopt an activated phenotype in many neurodegenerative diseases[2,3].

Genomic, ex vivo and preclinical data imply that microglial activation also may contribute to neurodegeneration[4], for example, by releasing inflammatory molecules in response to infectious or damage-related triggers[5]. These lead to both neuronal injury and, more directly,

pathological phagocytosis of synapses[5,6]. Development of tools which can reliably detect and quantify microglial activation in the living human brain has been an important goal. By enabling improved stratification and providing early pharmacodynamic readouts, these tools would accelerate experimental medicine studies probing disease mechanisms and early therapeutics.

Detection of 18 kDa Translocator Protein (TSPO) with positron emission tomography (PET) has been widely used to quantify microglial activation in vivo[7]. In the last 5 years alone, there have been ~300 clinical studies using TSPO PET to quantify microglial responses in the human brain, making it the most commonly used research imaging technique for this purpose. Such studies have examined TSPO in the context of neuroinflammatory disease, neurodegenerative disease, psychiatric conditions, and CNS tumours[8–11].

The TSPO signal is not specific to microglia, and the contribution from other cell types (particularly astrocytes and endothelial cells) is increasingly acknowledged[12]. The justification for quantifying TSPO as a marker of microglial activation is based on the assumption that when microglia become activated, they adopt a classical pro-inflammatory phenotype and TSPO expression is substantially increased[7,13,14]. This has been demonstrated repeatedly in mice, both in vitro and in vivo[15–18]. We have shown, however, that classical pro-inflammatory stimulation of human microglia and macrophages in vitro with the TLR4 ligand lipopolysaccharide (LPS) does not induce expression of TSPO[19]. Furthermore, in multiple sclerosis (MS), TSPO does not appear to be increased in microglia with activated morphology[20]. These data appear inconsistent with the assumption that TSPO is a marker of activated microglia in humans.

To address this issue, we performed a meta-analysis of publicly available expression array data and found that across a range of pro-inflammatory activation stimuli, TSPO expression is consistently and substantially increased in mouse, but not human macrophages and microglia in vitro. We then performed a comparative analysis of the TSPO promoter region in a range of mammalian species and found that the binding site for AP1 (a transcription factor which regulates macrophage activation in rodents[21]) is present in and unique to a subset of species within the *Muroidea* superfamily of rodents. Consistent with the hypothesis that this binding site is required for the increase in TSPO expression that accompanies pro-inflammatory stimulation, we show that TSPO is inducible by LPS in myeloid cells from the rat (another *Muroidea* species with the AP1 binding site in the TSPO core promoter) but not in other mammals. Because neuronal interactions modulate microglial phenotype, we then compared microglial TSPO expression in neurodegenerative diseases affecting the brain and spinal cord (Alzheimer's Disease (AD) and amyotrophic lateral sclerosis (ALS), respectively) as well as the classical neuroinflammatory brain disease MS which features highly activated microglia. We compared each human disease to its respective commonly used mouse model (amyloid precursor protein (*App*[NL-G-F])[22], tau (Tau[P301S])[23], superoxide dismutase 1 (SOD1[G93A])[24], and experimental autoimmune encephalomyelitis (EAE) in young and aged animals[25]. We also studied TSPO expression with EAE in the marmoset in conjunction with frequent MRI scanning that allowed for identification of the acute lesions which contain pro-inflammatory microglia. Consistent with the in vitro data, we show that in AD, ALS, and MS, and in marmoset EAE, TSPO protein expression does not increase in CNS myeloid cells that express a pro-inflammatory phenotype, while expression is markedly increased in activated myeloid cells in all mouse models of these diseases. With exploration of the relative expression of TSPO in publicly available CNS single-cell RNA sequencing (scRNAseq) data from brains of the human diseases and rodent models, we again show an increase in microglial TSPO gene expression in mice with pro-inflammatory stimuli, but not humans. Using functional studies and examination of transcriptomic co-expression networks, we find that TSPO is mechanistically linked to classical pro-inflammatory myeloid cell function in rodents but not

humans. Finally, by identifying genes with the AP1 binding site in their core promoter which are consistently upregulated in activated microglia in vitro and ex vivo, we show that TFEC and LCP2 are potential targets for a new generation of PET ligands, the binding of which could be sensitive to microglial activation state across a range of disease contexts in the human CNS.

These data suggest that the commonly held assumption that TSPO PET is sensitive to microglial activation is true only for a subset of species within the *Muroidea* superfamily of rodents. In contrast, in humans and other mammals, it simply reflects the local density of inflammatory cells irrespective of the disease context. The clinical interpretation of the TSPO PET signal therefore needs to be revised.

## Results

### *TSPO* expression and epigenetic regulation in primary macrophages

To investigate *TSPO* gene expression changes in human and mouse a meta-analysis was performed using publicly available macrophage and microglia transcriptomic datasets upon pro-inflammatory stimulation (Fig. 1). We found 10 datasets (Fig. 1a) derived from mouse macrophages and microglia in samples from 68 mice and with inflammatory stimuli including activation with LPS, Type 1 interferon (IFN), IFNγ, and LPS plus IFNγ. We performed a meta-analysis and found that *Tspo* was upregulated under pro-inflammatory conditions (Fig. 1a). In the individual datasets, *Tspo* was significantly upregulated in 9 of the 10 experiments. We then interrogated 42 datasets from primary human macrophages and microglia involving samples from 312 participants, with stimuli including inflammatory activation with LPS, IFNγ, IL1, IL6, PolyIC, viruses, and bacteria (Fig. 1b). In the meta-analysis, there was a non-significant trend towards a reduction in human *TSPO* expression under pro-inflammatory conditions (Fig. 1b). In the individual datasets, *TSPO* was unchanged in 33/42 (79%) of the datasets, significantly downregulated in 8/42 (19%) and significantly upregulated in 1/42 (2%). In contrast to the findings in mice, our analysis thus suggests that TSPO expression is not upregulated in human microglia and macrophages after pro-inflammatory stimulation in vitro.

Given that experimental conditions for these publicly available datasets differ, we performed a direct comparison of the effect of classical pro-inflammatory stimuli on human and mouse myeloid cells in vitro. We examined the effect of LPS (10 ng/mL and 100 ng/mL), IFNγ and IFNα (1 ng/mL and 10 ng/mL) at 4, 8 and 24 h on primary human monocyte-derived macrophages and primary mouse bone marrow-derived macrophages (BMDMs). Both human and mouse macrophages displayed evidence of activation as confirmed by TNFα release (for LPS and IFNγ) and IFI44 gene expression (for IFNα) (Fig. S1).

In BMDMs, LPS, IFNγ and IFNα all caused increases in *TSPO* gene expression which were most apparent at 8 h with the higher concentrations (Fig. 1c). However, there was no increase in *TSPO* gene expression in human macrophages under any of the conditions (Fig. 1d).

We examined the effect of LPS (10 ng/mL and 100 ng/mL), IFNγ and IFNα (1 ng/mL and 10 ng/mL) at 4, 8 and 24 h in primary mouse microglia. Primary human microglia were studied only with the high concentrations at the 24 h timepoint due to sample availability. Both human and mouse microglia displayed evidence of activation as confirmed by TNFα release (for LPS and IFNγ) and IFI44 gene expression (for IFNα) (Fig. S1). In mouse microglia, LPS, IFNγ and IFNα all caused increases in *TSPO* gene expression at the higher concentrations (Fig. 1e). However, in human microglia, TSPO gene expression did not change (Fig. 1f)

Rs6971 is a variant in the coding region of the *TSPO* gene which is common in humans but absent in rodents[26]. This variant leads to a single amino acid substitution which reduces the binding affinity of TSPO-targeting PET tracers for TSPO[26–28]. To determine whether the species-specific differences in pro-inflammatory induction of TSPO

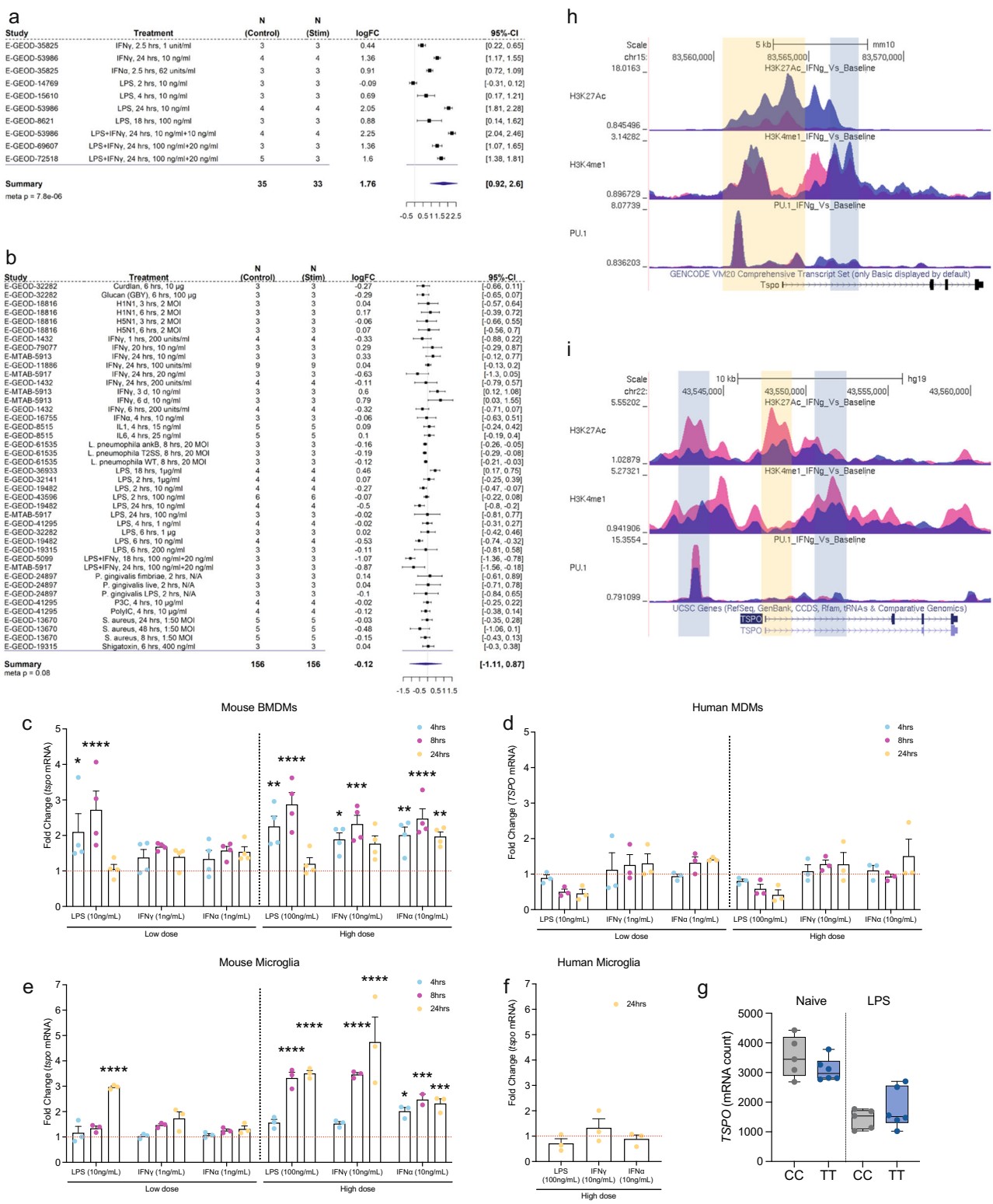

were due to the influence of rs6971, we activated primary human monocyte-derived macrophages from donors of different rs6971 genotype (C/C and T/T) with LPS (100 ng/mL for 24 h). TSPO expression did not increase in either genotype (Fig. 1g), ruling out rs6971 as an explanation for the species-specific differences in LPS induced TSPO expression. LPS induced a similar transcriptomic and cytokine response in both C/C and T/T individuals (Fig. S2).

To test whether TSPO gene expression changes are regulated at an epigenetic level, we analysed publicly available ChIP-seq datasets for

histone modification in mouse and human macrophages before and after treatment with IFNγ[29,30] (Fig. 1h, i). Levels of H3K27Ac and H3K4me1 histone marks in the promoter and enhancer regions are associated with increased gene expression[29,31]. While both histone modifications were increased after IFNγ treatment in TSPO promoter regions in macrophages from mouse, they were decreased in humans (Fig. 1h, i). Consistent with this epigenetic regulation, *Tspo* gene expression was upregulated in mouse macrophages after IFNγ but not in human macrophages in RNAseq data from the same set of samples (Fig. S3a).

**Fig. 1 | *TSPO* gene expression and epigenetic profile in human and mouse macrophages. a, b** Forest plot of the meta-analysis for TSPO expression in **a** mouse and **b** human myeloid cells treated with a pro-inflammatory stimulus. Statistical significance for individual dataset was done using linear model, the meta-analysis was performed using random-effect model (black square; logFC, horizontal lines; 95% CI, diamond; pooled logFC). **c, d** Fold change of TSPO mRNA in macrophages after stimulation indicating increases in *tspo* expression in mice but not in TSPO expression in humans. **e, f** An increase is observed in *tspo* expression after stimulation of mouse microglia but not in TSPO in human microglia. **g** TSPO mRNA count data from RNA sequencing of human monocytes isolated from healthy volunteers (with C/C and T/T genotype at the rs6971 locus) exposed to 100 ng/mL LPS for 24 h shows no effect of genotype on TSPO mRNA. **h, i** ChIP-seq data, generated from **h** mouse and **i** human myeloid cells treated with IFNγ, visualisation of histone modification peaks (H3K27Ac, K4me1) and PU.1 binding peaks at TSPO loci in IFNγ-treated (blue) and baseline (pink) conditions. Yellow vertical shading corresponds to the TSS along with promoter and light blue shading corresponds to the enhancer region of the loci. Biologically independent samples were used for all experiments (**c–f** *n* = 3 for all conditions, **g** *n* = 5 C/C and *n* = 6 T/T genotype). Statistical significance in (**c–g**) was determined by one-way ANOVA or Kruskal–Wallis test when not normally distributed or by a two-tailed unpaired *t*-test or two-tailed Mann–Whitney U-test when not normally distributed. Bar graphs indicate the mean ± SEM. Box and whiskers mark the 25th to 75th percentiles and min to max values, respectively, with the median indicated. Source data are provided as a Source data file.

The PU.1 transcription factor is a master regulator of macrophage proliferation and macrophage differentiation[32,33]. Because PU.1 increases *Tspo* gene expression in the immortalised C57/BL6 mouse microglia BV-2 cell line[34], we next investigated whether *TSPO* expression in macrophages is regulated by PU.1 binding in human in publicly available ChIP-seq datasets. An increase in PU.1 binding in the mouse *Tspo* promoter after IFNγ treatment was observed (Fig. 1h). However, PU.1 binding to the human *TSPO* promoter was decreased after IFNγ treatment (Fig. 1i). To test whether the reduced PU.1 binding at the human *TSPO* promoter was due to reduced PU.1 expression, we analysed RNAseq data from the same set of samples. Expression of SPI-1, the gene that codes for PU.1, was not altered in human macrophages after IFNγ treatment (Fig. S3b), suggesting that the reduced binding of PU.1 to the human *TSPO* promoter region was unlikely to be due to reduced PU.1 levels. This suggests that repressive chromatin remodelling in the human cells leads to decreased PU.1 binding, a consequence of which could be the downregulation of *TSPO* transcript expression. This is consistent with the meta-analysis (Fig. 1a, b); although *TSPO* expression with inflammatory stimuli did not significantly change in most studies, in 8/9 (89%) of studies where TSPO did significantly change, it was downregulated (Fig. 1b). Together this data shows that in vitro, pro-inflammatory stimulation of mouse myeloid cells increases TSPO expression, histone marks in the enhancer regions and PU.1 binding. These changes are not found following pro-inflammatory stimulation of human myeloid cells.

## The presence of the AP1 binding site in the TSPO promoter and LPS inducible TSPO expression is unique to the *Muroidea* superfamily of rodents

To understand why TSPO expression is inducible by pro-inflammatory stimuli in mouse but not human myeloid cells, we performed multiple sequence alignment of the *TSPO* promoter region of 15 species including primates, rodents, and other mammals (Fig. 2). We found that an AP1 binding site is present uniquely in a subset of species within the *Muroidea* superfamily of rodents including mouse, rat and chinese hamster (Fig. 2a). These binding sites were not present in other rodents (squirrel, guinea pig), nor in other non-rodent mammals (Fig. 2a). We generated a phylogenetic tree which shows a clear branching in the *TSPO* promoter of rat, mouse and chinese hamster from the other rodents and non-rodent mammals (Fig. 2b). Differential motif enrichment analysis of the *TSPO* promotor region between *Muroidea* vs non-*Muroidea* species confirmed a significant enrichment of the AP1 binding site in the *Muroidea* promoter (Fig. 2c). We expanded this motif search and *TSPO* promoter sequence divergence analysis to a wider range of 24 rodent species from the *Muroidea* superfamily and other non-*Muroidea* rodents. Again, we found that the AP1 site is confined only to a subset of the superfamily *Muroidea* (Fig. S4).

Silencing AP1 impairs LPS induced TSPO expression in the immortalized mouse BV2 cell line[34]. We therefore tested the hypothesis that LPS inducible TSPO expression occurs only in species with the AP1 binding site in the promoter region. In species that lack the AP1

binding site (human, pig, sheep, rabbit), TSPO expression was not induced by LPS (Fig. 2d). However, in the rat, where the AP1 binding site is present, TSPO was increased under these conditions (Fig. 2d).

## Microglial TSPO expression is unchanged in the AD hippocampus, but is increased in amyloid mouse models

Microglia-neuronal interactions, which modulate microglia inflammatory phenotype[1], are lost in monocultures in vitro. We therefore examined TSPO expression within inflammatory microglia in situ with quantitative neuropathology using *postmortem* samples from AD (Table S1). The specificity of the TSPO antibody to assess protein expression in these tissues was confirmed with a *Tspo* knock out mouse (Fig. S5). We compared immunofluorescence data from human *postmortem* AD brain to the *App*[NL-G-F] and TAU[P301S] mouse models.

We first examined the CA4 region of the hippocampus, one of the most severely affected regions in AD[35,36], comparing brain donors with severe disease (Braak 6) to controls with no neurological disease (Fig. 3a). No increases were observed in the number of IBA1+ microglia (Fig. 3b), nor GFAP+ astrocytes (Fig. 3c). Across the pan-microglial population, there was a 2-fold increase in HLA-DR signal intensity (Fig. 3d), providing evidence of myeloid cell activation as expected[37]. However, the density of TSPO+ cells in AD did not differ compared to controls (Fig. 3e) and the densities of TSPO+ microglia (Fig. 3f), TSPO + HLA-DR+ cells (Fig. 3g) and TSPO+ astrocytes (Fig. 3h) were similar for the two groups. We then quantified TSPO+ area (μm²) in microglia and astrocytes as an index of individual cellular expression (see 'Methods'). There was no difference in individual cellular TSPO expression in microglia or astrocytes (Fig. 3i) in AD relative to controls. In an independent cohort of AD donors (Braak 5/6, Table S2) and controls, we used confocal microscopy to generate high resolution z-stacked images of the CA4 region of the hippocampus. Again, we saw no changes in TSPO cellular expression in microglia in AD donors relative to control (Fig. 3j, k).

The closest ex vivo correlate of the in vivo PET signal is autoradiography. We therefore quantified the available binding sites for the TSPO radioligand [125I]CLINDE, in an independent cohort of hippocampal (CA4) brain samples from AD (Braak 5/6) and control donors (Table S3). As the binding affinity of [125I]CLINDE for TSPO depends on rs6971[38], we excluded rs6971 T/T samples (in whom minimal specific signal would be expected), and we controlled for C/T and C/C genotype using a fixed effects linear model. We found no difference in [125I]CLINDE specific binding in control samples relative to AD samples (Fig. S6). We also performed western blotting (WB) of TSPO protein density in a cohort of AD (Braak 5–6 vs Braak 0–2), which also showed no significant difference in the total tissue TSPO binding between the two groups (Fig. S7).

Comparing TSPO expression across the pan-microglial population in AD relative to control brains could potentially mask subtle differences which may be restricted to those microglia with a pro-inflammatory phenotype. We therefore next examined the expression of TSPO specifically in microglia in local pro-inflammatory contexts. We did this in two ways. First, we examined microglia that were

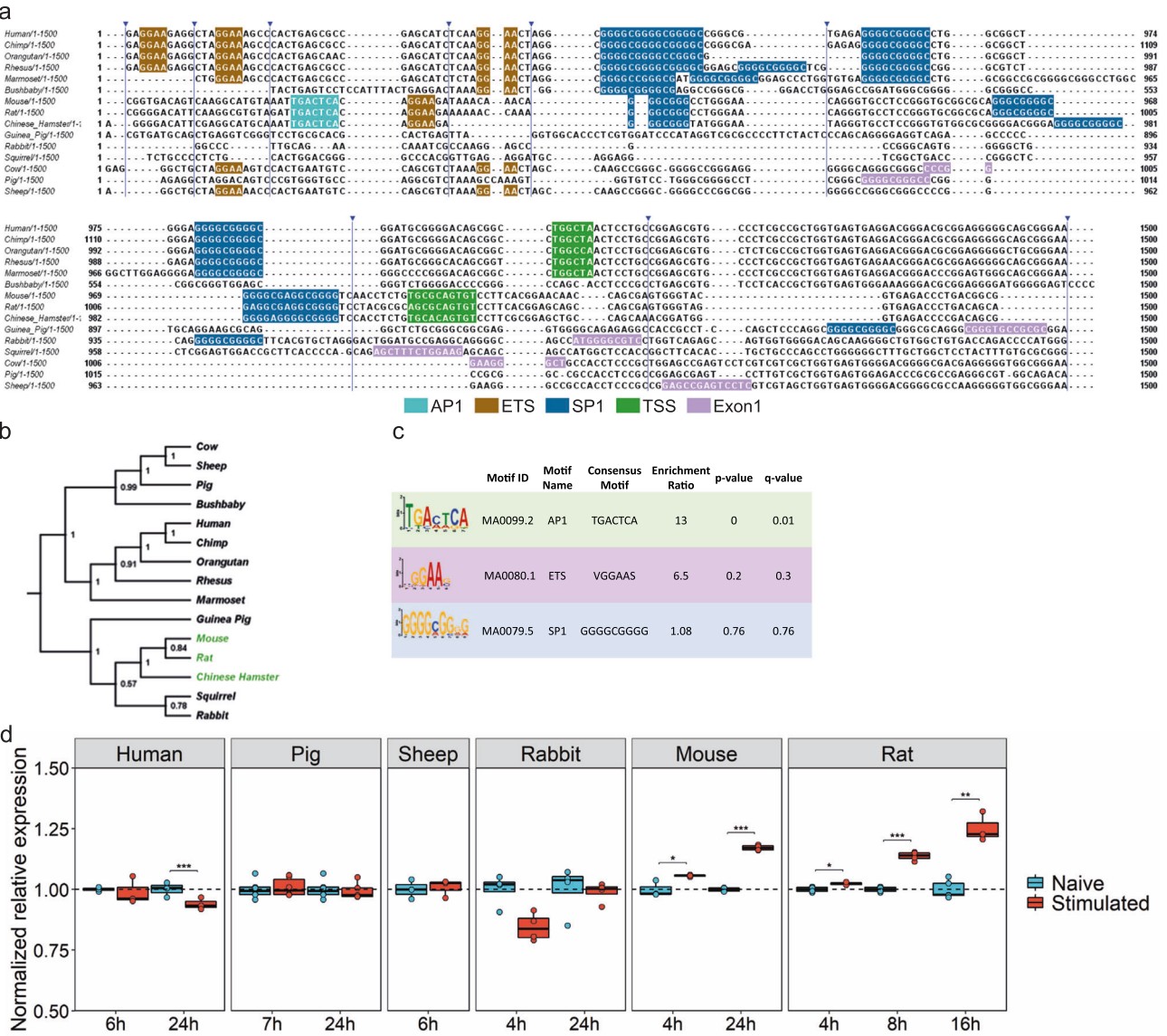

**Fig. 2 | AP1 binding site in the TSPO promoter and LPS inducible TSPO expression is unique to the Muroidea superfamily of rodents. a** Multiple sequence alignment of TSPO promoter region of 15 species from primate, rodent, non-primate mammals. AP1 (cyan) and an adjacent ETS (brown) site is present in only a sub-group of rodent family which includes mouse, rat, and Chinese hamster. The ETS site which binds transcription factor PU.1 is present across species. SP1 (blue) site is found in the core promoter close to the TSS (green). For species where the TSS is not known Exon1 (pink) location is shown. Blue arrowhead indicates sequence without any motif hidden for visualization. **b** Phylogenetic tree is showing a clear branching of rat, mouse, and Chinese hamster TSPO promoter from the rest of the species from rodents. Primates including marmoset forms a separate clade while sheep, cow and pig are part for the same branch. Green highlights represent species that contain the AP1 site in TSPO promoter. Phylogenetic tree was generated using the Maximum Parsimony method in MEGA11. The most parsimonious tree with length = 4279 is shown. The consistency index (CI) is 0.760458 (0.697014) and the retention index is 0.656386 (RI) (0.656386) for all sites and parsimony-informative sites (in parentheses). The percentage of replicate trees in which the

associated taxa clustered together in the bootstrap test (1000 replicates) are shown next to the branches. **c** Differential motif enrichment analysis between rodent vs non-rodent TSPO promoter region by SEA tools from MEME-suite confirms the significant enrichment of AP1 site in rodent promoter whereas SP1 site does not show any differential enrichment (Fisher's exact test was used to determine enrichment ratio and p-value, q-value was calculated by Benjamini & Hochberg method). TSS; Transcription start site. **d** TSPO gene expression in macrophages or microglia isolated from multiple species after LPS stimulation. In line with the multiple sequence alignment of the TSPO promoter, species (mouse, rat) that contains an adjacent AP1 and ETS (PU.1) motif shows an upregulation of TSPO gene after LPS stimulation. Species lacking (human, pig, sheep, rabbit) those sites show a downregulation or no change in expression after stimulation. Biologically independent samples were used for all experiments (**d** rabbit, rat, pig $n = 4$ and human, mouse, sheep $n = 3$ for all conditions). Box and whiskers mark the 25th to 75th percentiles and min to max values, respectively, with the median indicated. Source data are provided as a Source data file.

close to amyloid plaques or neurofibrillary tangles (NFTs) as these cells are activated relative to those microglia distant from the pathological features in brains with AD[37]. Second, we tested whether microglial TSPO expression was correlated with the intensity of classical markers of microglial activation. To address these questions, we studied an independent set of 22 AD brains (Braak 5/6, Medial Temporal Gyrus,

Table S4) and conducted multiplexed proteomics with imaging mass cytometry (IMC) to co-localise TSPO expression with markers of microglial phenotype, amyloid and tau (Fig. 4a). We also examined the previously described cohort (Table S2) with immunofluorescence and generated z-stacked images to assess TSPO expression in microglia close to and distant from amyloid plaques and NFTs.

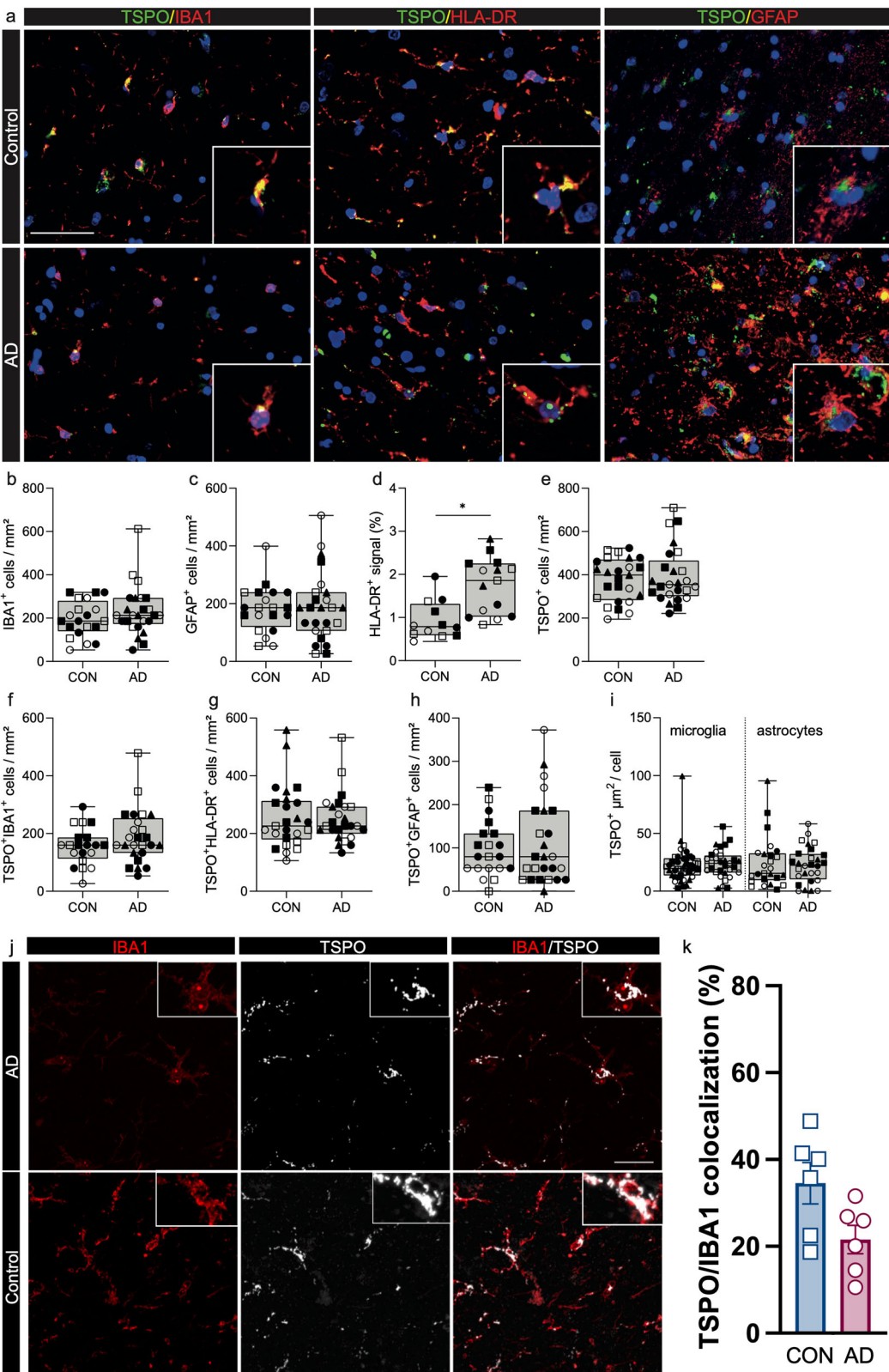

In the IMC cohort (Table S4) microglia within 10 μm of amyloid plaques contained 1.7 fold more amyloid than microglia far (>50 μm) from amyloid plaques (Fig. 4b). As expected from the literature[37], microglia close to plaques showed 1.4 and 2.1 fold increases in expression of activation markers HLA-DR and CD68, respectively, relative to microglia far from plaques (Fig. 4b). However, despite evidence of activation in microglia near to plaques, the expression of

TSPO did not differ from microglia far from plaques (Fig. 4b). We next examined microglia near NFTs. Microglia within 10 μm of NFTs contained 14-fold more tau deposits than microglia far (>50 μm) from NFTs (Fig. 4b), and showed a 3.5-fold increased expression of HLA-DR (Fig. 4b). However, again, despite this evidence of pro-inflammatory activation, TSPO expression in microglia close to NFTs did not differ from microglia far from NFTs (Fig. 4b). Consistent with our in vitro

**Fig. 3 | TSPO expression is not altered in the AD hippocampus.** Scale bar = 50 μm unless indicated, inserts are digitally zoomed in (200%). **a** Representative images of TSPO expression in microglia and astrocytes in AD hippocampus. **b**–**e** no increases were observed in microglia ($P = 0.5159$, U = 7, ranks = 17, 28) and astrocytes ($P = 0.8599$, $t = 0.1831$, df = 7). An increase was observed in HLA-DR+ signal in AD hippocampus. No increase was observed in TSPO+ cells ($P = 0.7329$, $t = 0.3534$, df = 8) in the AD hippocampus. **f**–**h** Concurrently no increases were observed in the number of TSPO + IBA1+ microglia ($P = 0.3573$, $t = 0.9854$, df = 7), TSPO + HLA-DR+ microglia ($P = 0.7239$, $t = 0.3659$, df = 8) and astrocytes ($P = 0.7181$, $t = 0.3760$, df = 7). **i** Even though microglia in the AD brain show signs of activation microglia do not upregulate TSPO expression in the hippocampus ($P = 0.6717$, $t = 0.4398$, df = 8), nor

do astrocytes ($P = 0.6475$, $t = 0.4750$, df = 8). **j** Representative images of TSPO expression in microglia in AD and control using confocal microscopy. Scale bar = 20 μm. **k** No changes were observed in TSPO cellular expression in microglia in AD donors ($n = 6$) relative to control ($n = 6$) using confocal imaging (Mann–Whitney U = 7, $P = 0.1775$). Biologically independent samples were used for all experiments (**b**, **c**, **f**, **h** $n = 4$ CON and $n = 5$ AD) (**d**, **e**, **g**, **i** $n = 5$ CON and 5 AD) (**k** $n = 6$ CON and $n = 6$ AD). Statistical significance in (**b**–**k**) was determined by a two-tailed unpaired $t$-test or two-tailed Mann–Whitney U-test when not normally distributed. Bar graphs mark the mean ± SEM. Box and whiskers mark the 25th to 75th percentiles and min to max values, respectively, with the median indicated. Each individual is represented by a different symbol.

data, TSPO genotype at rs6971 had no impact on microglial TSPO expression (statistics table).

We next correlated TSPO expression within microglia to expression of known markers of microglial activation (CD68 and HLA-DR). We found no correlation between TSPO and HLA-DR, and a significant but very weak correlation between TSPO and CD68 ($r^2 = 0.06$), again implying that TSPO does not increase when microglia are activated (Fig. S9).

We also examined the relationship between microglial TSPO expression and proximity to lesions with immunofluorescence. In the previously described AD cohort (Table S2) we labelled amyloid plaques, pTau, IBA1, and TSPO and generated z-stacked images (Fig. 4c). Again, we saw that microglia near to plaques and NFTs had no higher TSPO than microglia far from plaques and NFTs, respectively (Fig. 4d).

We next explored microglial TSPO gene expression in two independent AD datasets. We first examined a publicly available microglial scRNAseq dataset of AD (Braak III or below) and MCI brains *postmortem*[39]. Using AUCell to calculate the enrichment of microglial subclusters in pro-inflammatory, homeostatic and disease-associated[40] microglial gene sets, we identified pro-inflammatory and homeostatic microglial subclusters (Fig. S10). The pro-inflammatory subcluster showed upregulation of pro-inflammatory genes *APOE, HLA-DRA, TREM2, FTH1* and downregulation of the homeostatic *P2RY12* (Fig. 4e). However, TSPO expression was not increased in the pro-inflammatory subcluster relative to the homeostatic subcluster (Fig. 4e).

We also re-analysed a previously published snRNAseq dataset[41], including microglia nuclei enriched by selective depletion of nuclei from neurons and oligodendrocytes. TSPO expression, when measured in single nuclei, yields low counts. We therefore used a pseudobulking approach which bioinformatically transforms snRNAseq data to single cell type "bulk" RNAseq data by summing expression values over multiple nuclei per sample, and performs the analyses on individual samples rather than on individual nuclei. We compared the TSPO expression in pseudobulked data from a subcluster of activated microglia with a subcluster expressing predominantly genes associated with homeostatic microglia. As with the single cell dataset above, we confirmed that the pro-inflammatory subcluster showed upregulation of pro-inflammatory genes *APOE, HLA-DRA, TREM2, FTH1* and downregulation of the homeostatic *P2RY12* (Fig. 4f). Again, however, TSPO expression was not increased in the pro-inflammatory subcluster relative to the homeostatic subcluster (Fig. 4f). TSPO gene expression in nuclei from a brain region more severely affected by AD (entorhinal cortex) and from a brain region with a lower burden of AD pathology (somatosensory cortex) were compared and found to be similar in the two brain regions (statistics table). This further suggests that TSPO expression does not increase with the greater microglial activation associated with a greater burden of AD pathology. On the other hand, when we examined TSPO expression in vascular cells (Fig S11), we saw a ~3-fold increase in TSPO expression in AD relative to controls (statistics table), tentatively suggesting that increases in TSPO in vascular cells may contribute to the TSPO PET signal increase in AD.

We next compared the human AD data to that from mouse models $App^{NL-G-F}$ (Fig. 5a) and TAU$^{P301S}$ (Fig. 5h). The $App^{NL-G-F}$ model avoids artefacts introduced by APP overexpression by utilising a knock-in strategy to express human APP at wild-type levels and with appropriate cell-type and temporal specificity[22]. In this model, APP is not overexpressed. Instead, amyloid plaque density is elevated due to the combined effects of three mutations associated with familial AD (NL; Swedish, G: Arctic, F: Iberian). The $App^{NL-G-F}$ line is characterised by formation of amyloid plaques, microgliosis and astrocytosis[22]. We also investigated TSPO expression in a model of tauopathy, TAU$^{P301S}$ mice, which develop tangle-like inclusions in the brain parenchyma associated with microgliosis and astrocytosis[23]. The use of these two models allows differentiation of effects of the amyloid plaques and neurofibrillary tangles on the expression of TSPO in the mouse hippocampus. In $App^{NL-G-F}$ mice, an increase in the density of microglia was observed at 28 weeks (Fig. 5b), but not in the density of astrocytes (Fig. 5c). An increase in TSPO+ cells was also observed (Fig. 5d), due to an increase in numbers of TSPO+ microglia and macrophages (Fig. 5e). No differences were observed in the density of TSPO+ astrocytes in $App^{NL-G-F}$ at 10 weeks, although a small (relative to that with microglia) increase was observed at 28 weeks (Fig. 5f). Finally, we then quantified TSPO+ area in microglia and astrocytes as an index of TSPO expression in individual cells. In contrast to the human data, expression of TSPO in individual cells was increased by 3-fold in microglia in the $App^{NL-G-F}$ mice at 28 weeks (Fig. 5g). It was unchanged in astrocytes (Fig. 5g). In the TAU$^{P301S}$ mice, no differences were observed in microglia (Fig. 5i) or astrocyte (Fig. 5j) densities, in TSPO+ cell density (Fig. 5k), or in the density of TSPO+ microglia (Fig. 5l) or of TSPO+ astrocytes (Fig. 5m) in the hippocampus at either 8 or 20 weeks. However, as with the $App^{NL-G-F}$ mouse (and in contrast to the human), a 2-fold increase in individual cellular TSPO expression was observed within microglia in TAU$^{P301S}$ mice (Fig. 5n). Again, as with the $App^{NL-G-F}$ mouse, individual cellular TSPO expression within astrocytes was unchanged (Fig. 5n).

Finally, we examined a publicly available dataset of single-cell RNAseq in the 5XFAD mouse model of AD[40]. Disease-associated microglia (DAM) showed increased expression of classical markers including *Apoe, Trem2, Tyrobp* and *Cst7* (Fig. S12). Unlike in the human datasets, where the pro-inflammatory subclusters showed no change in TSPO expression, DAM microglial subclusters in the mouse model showed a significant upregulation of *TSPO* (Fig. S12).

Together, these data demonstrate that microglial expression of TSPO is increased with pathology and disease-related activation in $App^{NL-G-F}$ and TAU$^{P301S}$ mice, but not in human AD tissue. Furthermore, while TSPO gene expression is increased in DAM microglia found in mouse models, it was not increased in activated microglia from AD donors. TSPO gene and protein expression were unchanged in astrocytes in both mouse models and in the human disease.

## Microglial TSPO is upregulated in SOD1$^{G93A}$ mice but not in ALS

Spinal cord and brain microglia differ with respect to development, phenotype, and function[42]. We therefore next investigated ALS (Table S5), that primarily affects the spinal cord rather than the brain. We studied tissue blocks from human ALS donors that we had

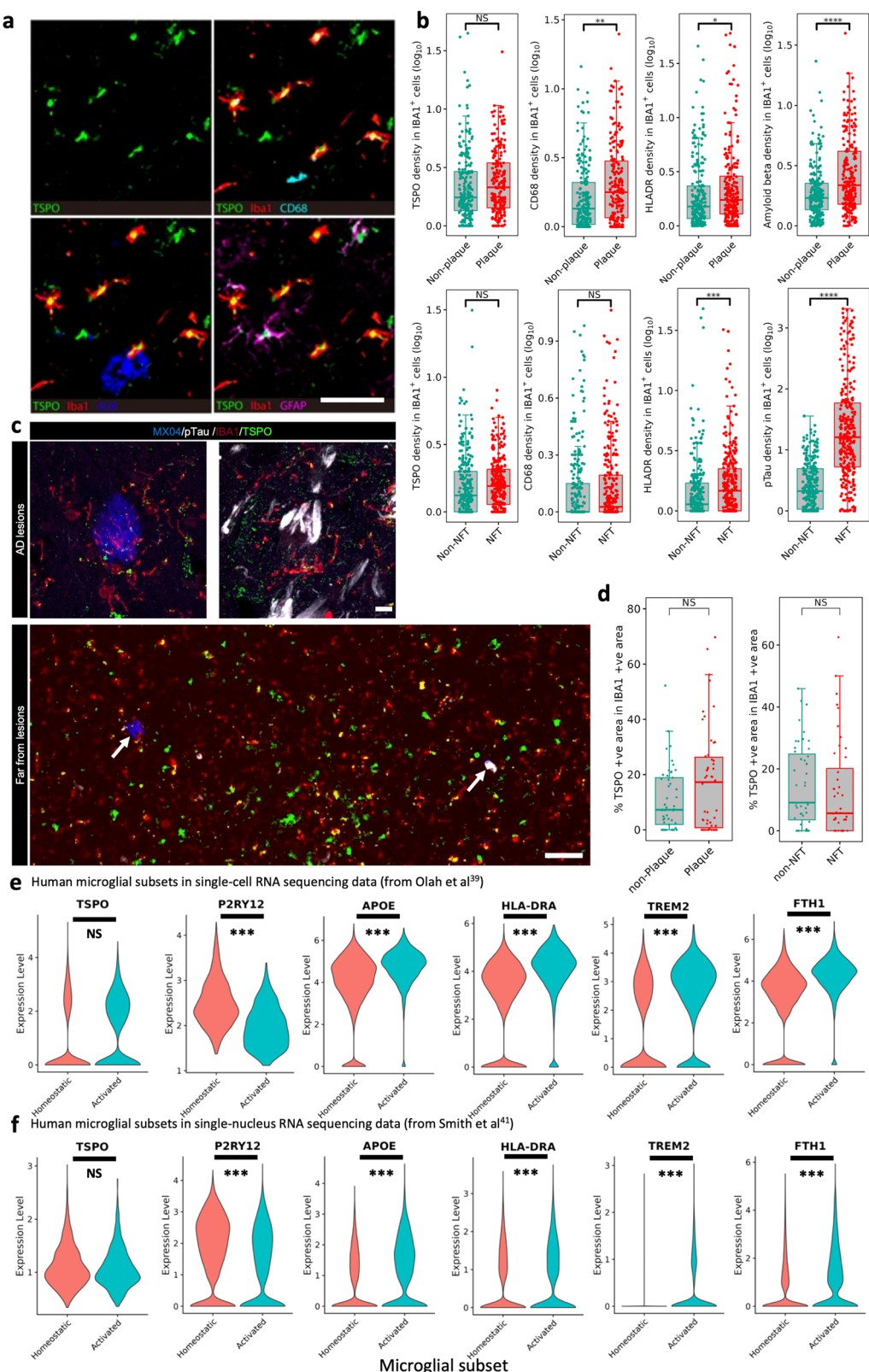

Human microglial subsets in single-cell RNA sequencing data (from Olah et al[39])

Human microglial subsets in single-nucleus RNA sequencing data (from Smith et al[41])

previously characterised and demonstrated TDP-43 pathology with a reduction in motor neuron density[43]. We compared this data to that from the commonly used SOD1[G93A] mouse model of ALS. TSPO expression in ALS was investigated in the ventral horn and lateral columns of the spinal cord in cervical, thoracic, and lumbar regions (Fig. 6a). An increase in microglia (Fig. 6b), HLA-DR+ microglia (Fig. 6c) and astrocytes (Fig. 6d) was observed in human ALS spinal cord. The

density of TSPO+ cells was increased by 2.5-fold in ALS spinal cords across all regions when compared to controls (Fig. 6e). No additional changes were found when stratifying the cohort based on disease duration or spinal cord regions, white or grey matter, or spinal cord levels. In comparison to the controls, ALS samples exhibited a 3-fold increase in the density of TSPO+ microglia (TSPO + IBA1+ cells, Fig. 6f) and a 3-fold increase in TSPO+ activated microglia/macrophages

**Fig. 4 | TSPO is not upregulated in human pro-inflammatory activated microglia. a** Representative IMC images of HLA-DR, CD68, GFAP, 4G8 and TSPO. **b** Using IMC TSPO is unaltered in IBA1+ cells in close proximity to amyloid plaques (*P* = 0.86), whereas CD68 and HLA-DR and beta amyloid are significantly upregulated (upper row, *P* = 0.0037, *P* = 0.037 and *P* = 2.97E−11, respectively). Similarly, TSPO is not upregulated in IBA1+ cells in close proximity to NFT (*P* = 0.809), whereas HLA-DR and pTau are significantly upregulated (lower row, *P* = 0.0002 and *P* = 3.25E−60, respectively). **c** Representative confocal microscopy images from an immunofluorescence experiment staining for beta amyloid plaques (blue), TSPO (green), IBA1 (red) and pTau (white). **d** Results of the quantification of z-stacked images from the immunofluorescence experiment from (**c**) showing that TSPO is not altered in IBA1+ cells in close proximity to amyloid plaques (*p* = 0.30) or NFT (*p* = 0.094). **e** In the pro-inflammatory microglial subcluster of the Olah et al. single-cell RNA sequencing dataset, there is downregulation of *P2RY12* (*P* = 2.29E−16) and upregulation of the pro-inflammatory *APOE* (*P* = 4.27E−104), *HLA-DRA* (*P* = 3.51E−77), *TREM2* (*P* = 1.80E−61) and *FTH1* (*P* = 6.51E−115) relative to the homeostatic microglial subcluster. However, TSPO expression is not upregulated (*P* = 0.99). **f** In the pro-inflammatory microglial subcluster of the Smith et al. single nucleus RNA sequencing dataset, there is downregulation of *P2RY12* (*P* = 3.58E−9) and upregulation of the pro-inflammatory *APOE* (*P* = 5.42E−12), *HLA-DRA* (*P* = 8.15E−5), *TREM2* (*P* = 1.18E−12) and *FTH1* (*P* = 8.11E−21) relative to the homeostatic microglial subcluster. However, TSPO expression is not upregulated (*P* = 0.44). Scale bar = 120 μm unless indicated. Biologically independent samples were used for all experiments (**b** for Plaques *n* = 446 cells from 22 individual samples, for NFTs *n* = 561 cells from 16 individual samples) (**d** *n* = 50 regions of interest for all conditions). Statistical significance in (**b**–**d**) was determined using a mixed-effects model and a zero-inflated Gamma distribution. Box and whiskers mark the 25th to 75th percentiles and the 95% confidence interval, respectively, with the median indicated. For demonstration purposes, the TSPO violin plot only contains the nuclei where TSPO is expressed, although as described in the methods the statistical analysis was performed on all nuclei. Source data are provided as a Source data file.

(TSPO + HLA-DR+ cells, Fig. 6g). A 2.5-fold increase in the density of TSPO+ astrocytes (TSPO + GFAP+ cells) was observed in ALS compared to control (Fig. 6h). We then quantified TSPO+ area in microglia and astrocytes as an index of individual cellular TSPO expression. No increase in TSPO+ area (μm²) was found in microglia or astrocytes in ALS when compared to control (Fig. 6i), implying that TSPO expression does not increase in microglia or astrocytes with ALS.

SOD1$^{G93A}$ mice express high levels of mutant SOD1 that initiates adult-onset neurodegeneration of spinal cord motor neurons leading to paralysis, and as such these mice have been used as a preclinical model for ALS[24]. To determine the extent to which TSPO+ cells were present in SOD1$^{G93A}$ mice TSPO+ microglia and astrocytes were quantified with immunofluorescence in the white and grey matter of the spinal cord (Fig. 6j). An increase was observed in the total number of microglia in 16-week-old SOD1$^{G93A}$ mice (Fig. 6k) and in astrocytes in 10- and 16-week-old animals (Fig. 6l). The density of TSPO+ cells was increased 2- to 3-fold in presymptomatic disease (10 weeks) compared to non-transgenic littermates in both white and grey matter (Fig. 6m). Increases in the density of TSPO + IBA1+ cells were not observed in SOD1$^{G93A}$ mice compared to control animals (Fig. 6n). However, a significant 8- to 15-fold increase in the density of TSPO + GFAP+ astrocytes was observed in 10- and 16-week-old SOD1$^{G93A}$ mice compared to 10- and 16-week-old wild-type mice (Fig. 6o). Finally, we then quantified TSPO+ area in microglia and astrocytes as an index of individual cellular TSPO expression. In contrast to the human data, where there was no change in disease samples relative to controls, expression of TSPO in individual cells was increased by 1.5-fold in microglia in the rodent model (Fig. 6p). As with the *App*$^{NL-G-F}$ and TAU$^{P301S}$ mice above, TSPO expression within astrocytes was unchanged (Fig. 6p).

In summary, consistent with the data from AD and relevant mouse models, we have shown that TSPO expression is increased within microglia from SOD1$^{G93A}$ mice, but not increased in microglia from human ALS tissue. TSPO also was unchanged in astrocytes from the SOD1$^{G93A}$ mice and the human disease relatively to those in the healthy control tissues.

## Increased myeloid cell TSPO expression is found in mouse EAE, but not in MS or marmoset EAE

Having found no evidence of increased TSPO expression in activated microglia in human neurodegenerative diseases affecting the brain or spinal cord, we next examined MS as an example of a classical neuroinflammatory disease characterised by microglia with a highly activated pro-inflammatory phenotype. We compared data from human *postmortem* MS brain (Table S6) to mice with EAE (Table S7). We also examined brain tissue from marmoset EAE (Table S8), as *antemortem* MRI assessments in these animals allow for identification of acute lesions which are highly inflammatory.

We previously defined TSPO cellular expression in MS[20,44]. HLA-DR+ microglia expressing TSPO were increased up to 14-fold in active lesions compared to white matter from control donors[44], and these microglia colocalised with CD68 and had lost homeostatic markers P2RY12 and TMEM119, indicating an activated microglial state[20]. Here we quantified individual cellular TSPO expression in both microglia and astrocytes by comparing cells in active white matter lesions to cells from both normal appearing white matter (NAWM) from the same donors, and from white matter from control donors. Consistent with the human data from AD and ALS, there was no difference in TSPO expression in individual microglia or astrocytes in active white matter lesions in MS compared to NAWM in MS or control tissue (Fig. 7a, b).

We next investigated the relative levels of TSPO expression (Fig. 7c) in microglia and astrocytes in EAE, a commonly used experimental mouse model of MS in which acute disease (aEAE), remissions and chronic relapsing disease (CREAE) is induced in Biozzi ABH mice[25,45]. Neurodegenerative diseases typically occur in old age, whereas aEAE, CREAE and the AD and ALS relevant rodent models described above are induced in young mice. As age might affect TSPO regulation[46], we also investigated TSPO expression in progressive EAE (PEAE), a model where the acute neurological disease and corresponding pathology is induced in aged mice (12 months).

Increases in numbers of both microglia and astrocytes were observed in aEAE as well as in PEAE mice compared to their respective young and old control groups (Fig. 7d, e). Similarly, increases were observed in the number of TSPO⁺ cells, TSPO+ microglia and TSPO+ astrocytes in both aEAE and PEAE relative to their respective controls (Fig. 7f–h). When comparing the young control mice (aEAE controls) with the old control mice (PEAE controls), no differences were observed in microglial and TSPO+ microglial density (Fig. 7d, g). Similarly, there was no difference in density of astrocytes or TSPO+ astrocytes between these two control groups (Fig. 7e, h).

To investigate individual cellular TSPO expression, TSPO+ area was measured in microglia and astrocytes. Individual microglia expressed 3-fold greater TSPO and 2-fold greater TSPO in aEAE and PEAE, respectively, relative to their control groups (Fig. 7i, j). The individual cellular TSPO expression was not higher in microglia from young mice relative to old mice (Fig. 7i, j). Again, as with the SOD1$^{G93A}$, *App*$^{NL-G-F}$, and TAU$^{P301S}$ mice, individual cellular TSPO expression within astrocytes was unchanged (Fig. 7i, j).

Finally, we investigated TSPO expression in EAE induced in the common marmoset (*Callithrix jacchus*) (Fig. S13), a non-human primate which, like humans, lacks the AP1 binding site in the core promoter region of TSPO. Both the neural architecture and the immune system of the marmoset are more similar to humans than those of the mouse[47–49]. Marmoset EAE therefore has features of the human disease

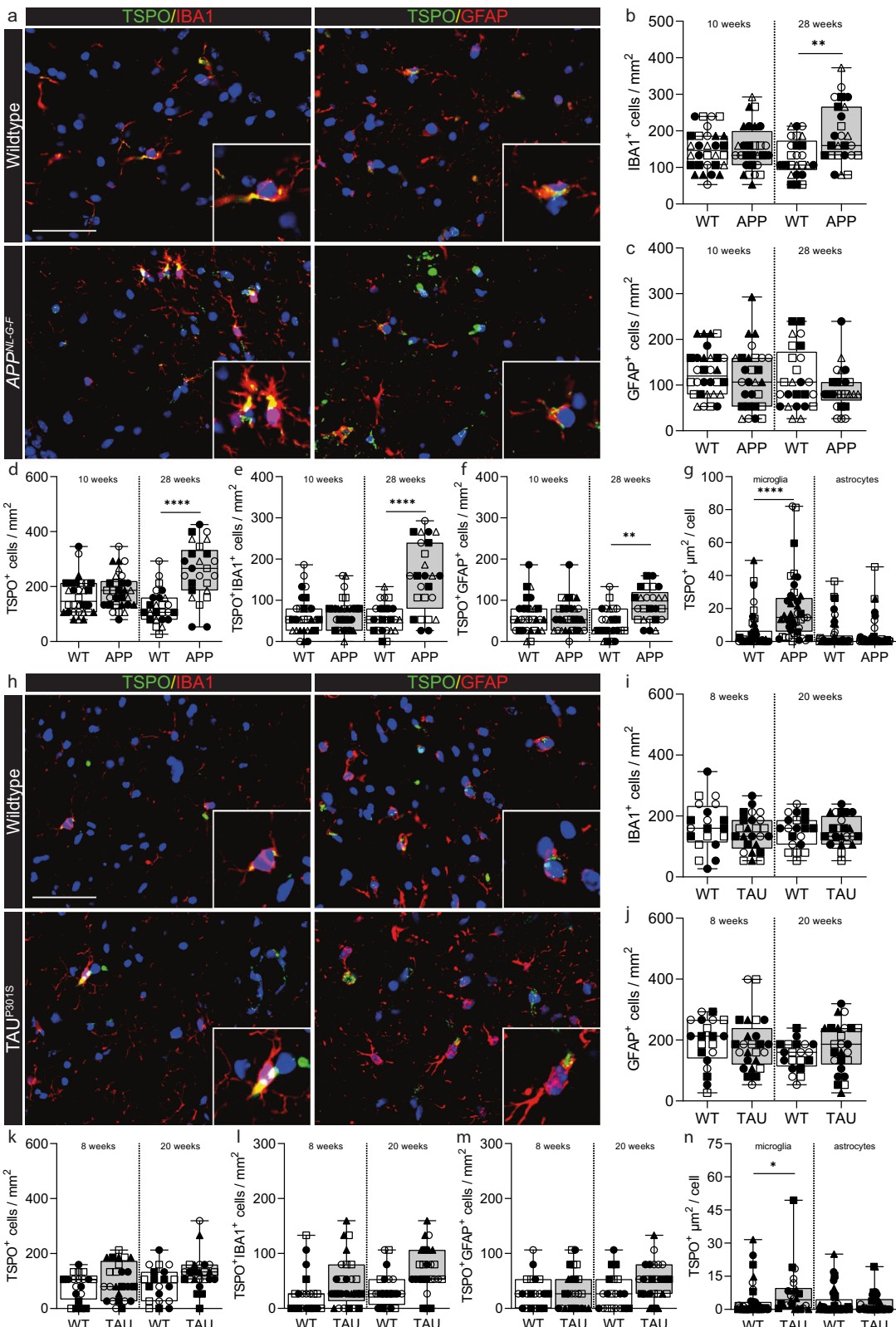

which are not seen in mouse EAE, such as perivenular white matter lesions identifiable by MRI. Marmosets were scanned with MRI biweekly, which allowed the ages of lesions to be determined and the identification of acute lesions including pro-inflammatory microglia. In acute and subacute lesions, there was an increase of up to 27-fold in the density of TSPO+ microglia relative to white matter from controls (Fig. S13a–c) and these microglia bore the hallmarks of pro-

inflammatory activation. However, TSPO expression in individual microglia, here defined as the percentage of TSPO+ pixels using immunofluorescence, was not increased in acute or subacute lesions relative to control (Fig. S13d).

In summary, and consistent with the AD and ALS data, we have shown that individual cellular TSPO expression is increased in microglia in EAE in both young and aged mouse models, but it is not

**Fig. 5 | Microglia in the $App^{NL-G-F}$ and TAU$^{P301S}$ model increase TSPO expression.**
**a** Representative images of TSPO expression in microglia and astrocytes in $App^{NL-G-F}$ hippocampus. **b** An increase was observed in IBA1+ microglia at 28 weeks ($P = 0.0078$, $t = 3.522$, df = 8) but not 10 weeks ($P = 0.8788$, $t = 0.1565$, df = 10) in $App^{NL-G-F}$ hippocampus compared to control. **c** No increase in astrocytes was observed (10 weeks: $P = 0.6266$, $t = 0.5019$, df = 10; 28 weeks: $P = 0.4425$, $t = 0.8080$, df = 8). **d** TSPO+ cells were increased at 28 weeks ($P = 0.0079$, U = 0, ranks = 15, 40) but not at 10 weeks ($P = 0.2375$, $t = 1.257$, df = 10) in the $App^{NL-G-F}$ mice. **e, f** Both TSPO+ microglia ($P = 0.0005$, $t = 5.658$, df = 8) and astrocytes ($P = 0.0030$, $t = 4.207$, df = 8) were increased at 28 weeks in the hippocampus of $App^{NL-G-F}$ mice but not at 10 weeks (microglia: $P = 0.7213$, $t = 0.3670$, df = 10; astrocytes: $P = 0.9561$, $t = 0.056$, df = 10). **g** Activated microglia ($P < 0.0001$, $t = 7.925$, df = 8), but not astrocytes ($P = 0.3095$, U = 7, ranks = 33, 22), in the $App^{NL-G-F}$ model have increased TSPO expression at 28 weeks. **h** Representative images of TSPO expression in microglia and astrocytes in TAU$^{P301S}$ hippocampus. **i–k** No increases in microglia (8 weeks: $P = 0.3687$, $t = 0.9608$, df = 7; 20 weeks; $P = 0.9647$, $t = 0.04580$, df = 7), astrocytes

(8 weeks: $P = 0.7353$, $t = 0.3519$, df = 7; 20 weeks; $P = 0.0870$, $t = 1.989$, df = 7) or TSPO+ cells (8 weeks: $P = 0.8492$, U = 9, ranks = 19, 26; 20 weeks; $P = 0.0876$, $t = 1.985$, df = 7) were observed in the hippocampus of TAU$^{P301S}$ mice. **l, m** No increase was observed in the number of TSPO+ microglia (8 weeks: $P = 0.2787$, $t = 1.174$, df = 7; 20 weeks; $P = 0.0907$, $t = 1.961$, df = 7) or astrocytes (8 weeks: $P = 0.8684$, $t = 0.1718$, df = 7; 20 weeks: $P = 0.1984$, U = 4.5, ranks = 14.5, 30.5). **n** Microglia in the TAU$^{P301S}$ increase TSPO expression ($P = 0.0133$, $t = 3.471$, df = 6) whereas astrocytes do not ($P = 0.5800$, $t = 0.5849$, df = 6). Biologically independent samples were used for all experiments (**b–g** 10 weeks: $n = 6$ WT and $n = 6$ $App^{NL-G-F}$, 28 weeks: $n = 5$ WT and $n = 5$ $App^{NL-G-F}$) (**i–n** 8 and 20 weeks: $n = 4$ WT and $n = 5$ TAU$^{P301S}$). Statistical significance in (**b–g**) and (**i–n**) was determined by a two-tailed unpaired $t$-test or two-tailed Mann–Whitney U-test when not normally distributed. Box and whiskers mark the 25th to 75th percentiles and min to max values, respectively, with the median indicated. Scale bar = 50 μm, inserts are digitally zoomed in (200%). Each individual is represented by a different symbol. Source data are provided as a Source data file.

increased in microglia from MS lesions nor marmoset EAE acute lesions. Again, consistent with previous data, astrocytes did not show an increase in TSPO expression in either MS or EAE.

### TSPO is mechanistically linked to classical pro-inflammatory myeloid cell function in mice but not humans

Having demonstrated species-specific differences in TSPO expression and regulation, we then sought to examine TSPO function in mouse and human myeloid cells. We first examined the effect of pharmacological modulation of the classical microglial pro-inflammatory phenotype using the high-affinity TSPO ligand, XBD173. Consistent with the literature[15–17], we found that in primary mouse macrophages and the BV2 mouse microglial cell line, XBD173 reduced LPS induced release of pro-inflammatory cytokines (Fig. 8a–c). However, in primary human macrophages from rs6971 C/C or T/T donors and in human induced pluripotent stem cell (hIPSC) derived microglia, XBD173 had no impact on the release of these cytokines, even at high concentrations associated with 98% TSPO binding site occupancy (Fig. 8d–g). We found similar results for zymosan phagocytosis. Primary mouse microglia demonstrated a dose dependent increase in phagocytosis upon exposure to XBD173 (Fig. 8h). However, we saw no increase in phagocytosis in primary human macrophages upon XBD173 exposure (Fig. 8i).

XBD173 is metabolised by CYP3A4, which is expressed in myeloid cells. We therefore used LC-MSMS to quantify XBD173 in the supernatant to test the hypothesis that the lack of drug effect on human myeloid cells was due to depletion of XBD173. The measured concentration of XBD173 in the supernatant at the end of the assay was no different to the planned concentration (Fig. S14), excluding the possibility that XBD173 metabolism explained the lack of effect.

To understand if TSPO is associated with divergent functional modules in mouse and human we then used weighted gene co-expression network analysis to examine the genes whose expression are correlated with *TSPO* in mouse and human myeloid cells. To construct the gene co-expression networks, we used four publicly available and one in-house RNA-seq data from human[50–53] ($n = 47$) and five publicly available mouse[54–57] ($n = 35$) datasets of myeloid cells treated with LPS or LPS and IFNγ. In mouse myeloid cells, the gene ontology biological processes associated with the TSPO network related to classical pro-inflammatory functions such as responses to type 1 and 2 interferons, viruses, and regulation of cytokine production (Fig. 8j, Supplementary Data File 1). However, in human myeloid cells, the processes associated with the TSPO co-expression network related to bioenergetic functions such as ATP hydrolysis, respiratory chain complex assembly, and proton transport (Fig. 8k, Supplementary Data File 1). There was no overlap in the genes that TSPO is co-expressed with in mouse, relative to human, myeloid cells (Fig. 8l).

### TFEC and LCP2 may offer alternative targets to detect microglial activation

Together, these data suggest TSPO is a marker of activated microglia in rodents but not humans because it is under AP1 control in the former but not the latter. The expression of TSPO therefore cannot be used as an index of microglial activation in humans. However, identifying markers specific for activated microglia is an important goal for both clinical fluid and imaging biomarker research. To identify new potential markers specific for activated microglia, we identified genes which have the AP1 binding site in their core promoter region and tested for their upregulation with inflammatory stimuli in a meta-analysis (also used to generate data for Fig. 1), which featured 42 experiments involving a range of pro-inflammatory activation stimuli with human myeloid cells in vitro. 96 genes met these two criteria (Table S10). We tested for upregulation of the 96 genes in microglia from snRNAseq datasets of AD[41] and MS[58] and in bulk RNAseq data in ALS as we are not aware of publicly available snRNAseq from spinal cord of donors with this disease[59]. Of the 96 genes with AP1 binding sites which were upregulated in microglia and macrophages with inflammatory stimuli in vitro, only Hexokinase 2 (HK2), Lymphocyte Cytosolic Protein 2 (LCP2) and Transcription Factor EC (TFEC) were upregulated in AD, MS, and ALS relative to non-disease controls. We tested each gene for CNS cell type specificity and found that TFEC and LCP2 are both highly selective for microglia (Fig. S15). Both TFEC and LCP2 showed greater selectivity for microglia than did TSPO, which was highly expressed in astrocytes and endothelial cells as well as microglia (Fig. S15).

We then investigated TFEC and LCP2 gene expression in human macrophages in vitro. TFEC was increased 4-, 4- and 6-fold, respectively, following 24 h stimulation with LPS (100 ng/mL), IFNg (10 ng/mL) or IFNa (10 ng/mL) (Fig. S16). LCP2 was increased 3-, 4- and 6-fold, respectively, following the same activation protocols (Fig. S16). Finally, we examined protein expression of LCP2 in human brain *postmortem* (we were unable to find a TFEC antibody which produced a measurable signal in the CNS). We found that microglial LCP2 was upregulated by 10-, 10- and 2-fold from AD, MS and ALS donors, respectively, relative to control (Figs. S17, S18). Where immunofluorescence was used (MS, ALS), we were also able to test for changes in LCP2 expression per microglia. In MS active lesions relative to NAWM and ALS relative to control, LCP2 expression within microglia was increased by 2.8- and 1.3-fold, respectively (statistics table). We propose TFEC and LCP2 as candidate biomarkers of microglial activation in a range of disease contexts.

### Discussion

Microglial activation accompanies, and is a major contributor to, neurodegenerative and neuroinflammatory diseases[1,4–6,60]. A better understanding of microglial activation in combination with a technique that could reliably quantify activated microglia in the human CNS

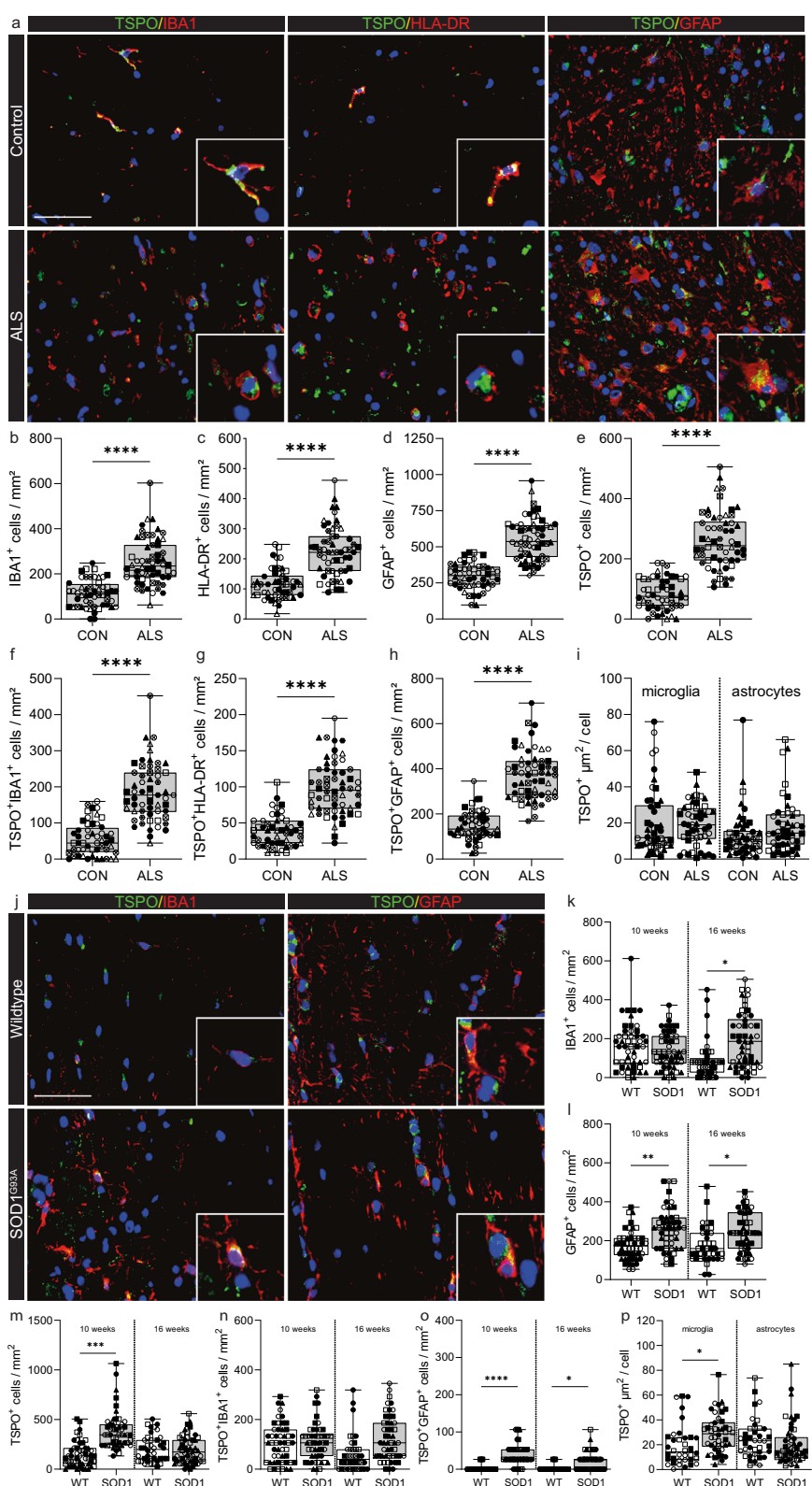

would have broad utility to monitor disease progression as well as response to therapy. TSPO PET has been applied by many with this objective[13,14]. Here we have tested the widely held assumption that *TSPO* cellular expression increases upon microglial activation. We examined in vitro data from isolated myeloid cells across 6 species, multiple sequence alignment of the TSPO promoter region across 34 species, and ex vivo neuropathological and scRNAseq data from human neuroinflammatory and neurodegenerative diseases, with

relevant marmoset and young and aged mouse models. We show that TSPO expression increases in mouse and rat microglia when they are activated by a range of stimuli, but that this phenomenon is unique to microglia from a subset of species from the *Muroidea* superfamily of rodents. The increase in TSPO expression is likely dependent on the AP1 binding site in the core promoter region of TSPO. We showed that TSPO is mechanistically linked to classical pro-inflammatory myeloid cell function in mice but not humans. Finally, we show that TFEC and

**Fig. 6 | TSPO is increased in microglia in SOD1$^{G93A}$ mice but not in ALS spinal cord. a** Representative images of TSPO expression in microglia and astrocytes in ALS spinal cord. **b–d** An increase was observed in microglia ($P < 0.0001$, $t = 7.445$, df = 19), HLA-DR+ microglia ($P < 0.0001$, $t = 6.007$, df = 19), and astrocytes ($P < 0.0001$, $t = 9.024$, df = 19) in ALS spinal cord when compared to controls. **e** A 2.5-fold increase of TSPO+ cells ($P < 0.0001$, $t = 12.88$, df = 19) was observed in the ALS spinal cord. **f, g** Up to a 3.4-fold increase in the density of TSPO+ microglia (TSPO + IBA1+ cell, $P < 0.0001$, $t = 7.541$, df = 19) (TSPO + HLA-DR+ cells, $P < 0.0001$, $t = 3.368$, df = 19) was observed. **h** TSPO+ astrocytes were significantly increased ($P < 0.0001$, $t = 11.77$, df = 19) in the spinal cord of ALS patients. **i** The increase in activated microglia and astrocytes was not associated with an increase in TSPO expression in microglia ($P = 0.7684$, $t = 0.3046$, df = 8) or in astrocytes ($P = 0.5047$, $t = 0.6985$, df = 8). **j** Representative images of TSPO expression in microglia and astrocytes in SOD1$^{G93A}$ spinal cord. **k** An increase was observed in microglia in SOD1$^{G93A}$ spinal cord when compared to controls at 16 weeks ($P = 0.0115$, $t = 3.395$, df = 7) but not at 10 weeks ($P = 0.5334$, $t = 0.6509$, df = 8). **l** An increase for astrocytes was observed for both 10 weeks ($P = 0.0024$, $t = 4.362$, df = 8) and 16 weeks ($P = 0.0248$, $t = 2.848$, df = 7). **m** An increase in TSPO+ cells was observed at 10 weeks ($P = 0.0011$, $t = 4.931$, df = 8) but not 16 weeks ($P = 0.7299$, $t = 0.3594$, df = 7). **n** No increase in the number of TSPO+ microglia was observed (10 weeks: $P = 0.5244$, $t = 0.6656$, df = 8; 16 weeks, $P = 0.0930$, $t = 1.944$, df = 7). **o** TSPO+ astrocytes were increased up to 15-fold in the spinal cord of SOD1$^{G93A}$ mice (10 weeks: $P = 0.0003$, $t = 6.085$, df = 8; 16 weeks: $P = 0.382$, $t = 2.548$, df = 7). **p** Despite no increase in the number of TSPO+ microglia, an increase in the amount of TSPO per cell was observed in microglia ($P = 0.0451$, $t = 2.435$, df = 7), but not astrocytes ($P = 0.4052$, $t = 0.8856$, df = 7) at 16 weeks. Biologically independent samples were used for all experiments (**b–h** $n = 10$ CON and $n = 11$ ALS) (**i** $n = 5$ CON and $n = 5$ ALS) (**k–p** 10 weeks: $n = 5$ WT and $n = 5$ SOD1$^{G93A}$, 16 weeks: $n = 4$ WT and $n = 5$ SOD1$^{G93A}$). Statistical significance in **b–i**, and **k–p** was determined by a two-tailed unpaired $t$-test. Box and whiskers mark the 25th to 75th percentiles and min to max values, respectively, with the median indicated. Scale bar = 50 μm, inserts are digitally zoomed in (200%). Source data are provided as a Source data file.

LCP2 are both selective for microglia and consistently upregulated in activated microglia in vitro and ex vivo, and hence are potential targets to quantify microglial activation in a range of contexts in the human brain.

This finding fundamentally alters the way in which the TSPO PET signal is interpreted, because it implies that the microglial component of the TSPO PET signal reflects density only, rather than a composite of density and activation phenotype. The inflammatory neuropil is often charactered by increases in microglial density and hence our data does not imply TSPO PET lacks utility. However, microglial activation can occur independently of increases in density. For example, in Parkinson's Disease (PD) there is evidence of activated microglia in the *postmortem* brain but minimal change in microglial density[61]. Three well-designed studies using modern TSPO radiotracers found no difference in TSPO signal between PD and controls groups[62–64]. The lack of increase in the TSPO PET signal is consistent with the data presented here, and should therefore not be interpreted as evidence for lack of microglial activation in PD. There also are situations in which microglial density is increased, but the microglia do not show evidence for pro-inflammatory activation, e.g., in rims of chronic active lesions with MS[58]. Our data also has implications for mechanistic investigations of TSPO function within microglia[65], and emphasises that such studies should include human-derived cells.

The data presented here also has implications for the development of novel PET tracers for microglial activation, suggesting that regulation by AP1 may be a characteristic of microglial proteins whose expression is correlated with cellular activation across a range of contexts. Using this approach, we identified LCP2 and TFEC as putative markers of microglial activation. Both proteins were also highly selective for microglia in the human brain. Of note, they both show much greater microglial selectivity than does TSPO, which is expressed across a range of cell types including astrocytes and endothelial cells. As has been reviewed elsewhere[66], the lack of microglial specificity for TSPO further complicates interpretation of the TSPO PET signal, as the source of the increased signal will be partly determined by the insult and ensemble of cells responding to it.

Our study has several limitations. First, we have only examined microglia under certain pro-inflammatory conditions and cannot exclude the possibility that other stimulation paradigms would increase TSPO in human myeloid cells. However, the in vitro stimuli which were examined included a broad range of pro-inflammatory triggers, and the three human diseases are diverse with respect to the mechanisms underlying the activation of microglia. Second, IF measurements are semi-quantitative. However, the same IF quantification methods were used in all human and mouse comparisons, and these methods consistently detected cellular TSPO increases in mouse microglia despite not detecting analogous changes in human microglia. Furthermore, where IMC and autoradiography was used, the quantitative data were consistent with IF. The neuropathology protein quantification was also consistent with gene expression measured by pseudobulking of enriched snRNAseq data. Third, for RNAseq analyses, we were restricted to scRNAseq and pseudobulking of snRNAseq data from nuclei enriched for microglia, as in conventional snRNASeq data *TSPO* is detected in only 5–12% of microglial nuclei[41,67–69]. Fourth, we chose three exemplar diseases with their respective animal models to span microglial activation across a range of contexts, but we did not include an analysis of a psychiatric disorder. Although we saw no change in microglial TSPO gene expression in bipolar disorder (Fig. S21, Table S11), future work should examine TSPO expression in psychiatric disorders in more detail. Finally, whilst we present data correlating inducible TSPO expression with the presence of the AP1 binding site in the TSPO core promoter region, to demonstrate causation, the AP1 binding site would need to be knocked out from the mouse or rat, and knocked in to a non-*Muroidea* rodent. Furthermore, although we were able to find array expression data for a range of non-rodent mammals that show TSPO is not induced upon myeloid cell activation, we were unable to find array expression data for those rodents that lack the AP1 binding site, such as squirrel or naked mole rat.

In summary, we present in vitro expression and sequence alignment data from a range of species, as well as ex vivo data from neurodegenerative and neuroinflammatory diseases and associated animal models. We show that inflammation-induced increases in cellular TSPO expression are restricted to microglia from a subset of species within the *Muroidea* superfamily of rodents, and that TSPO is mechanistically linked to classical pro-inflammatory myeloid cell function in mice, but not humans. The TSPO PET signal is best interpreted as reflecting the local density of inflammatory cells irrespective of phenotype; TSPO PET from humans cannot be interpreted as a measure of microglial activation as in mice or rats. Future work should explore new candidate biomarkers of microglial activation such as TFEC or LCP2.

## Methods
### Ethical oversight
All animal procedures have complied with their local and institutional guidelines. SOD1G93A mice were obtained in Italy and all procedures involving animals and their care were conducted in conformity with the following laws, regulations, and policies governing the care and use of laboratory animals: Italian Governing Law (D.lgs 26/2014; Authorization 19/2008-A issued 6 March, 2008 by Ministry of Health); Mario Negri Institutional Regulations and Policies providing internal authorization for persons conducting animal experiments; the National Institutes of Health's Guide for the Care and Use of Laboratory Animals (2011 edition), and European Union directives and guidelines (EEC Council Directive, 63/2010/EU). Studies with SOD1G93A mice

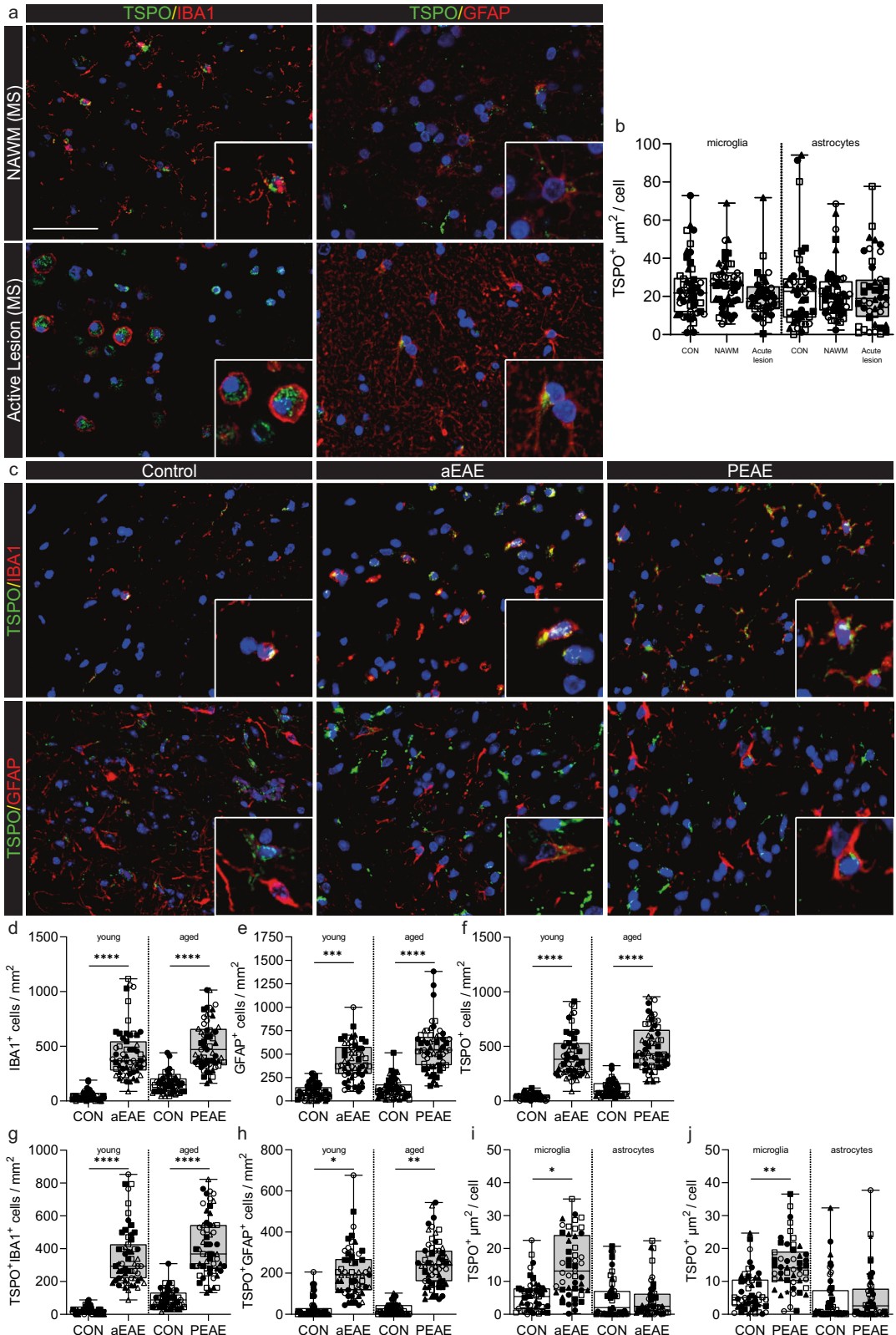

were approved by the Istituto di Ricerche Farmacologiche Mario Negri Animal Care and Use Committee (ref.nr. 566/2017PR). EAE mice, APPNLGF mice, TAUP301S mice were obtained in the United Kingdom and all procedures involving animals and their care were conducted in conformity with the following laws, regulations, and policies governing the care and use of laboratory animals: National guidelines (UK Animals Scientific Procedures Act 1986 and The Animal Welfare Act 2006). Animal studies were approved by institutional RECs. Studies with EAE mice were approved by the Queen Mary University of London Animal Welfare and Ethical Review Body and the United Kingdom Government Home Office Inspectorate (ref.nr. PPL 70/8699). Studies with APPNLGF mice were approved and covered by Imperial College's Animal Welfare and Ethical Review Body (AWERB) and Home Office Procedure Project Licence (PPL) (P8A2343FE). Studies with TAUP301S

**Fig. 7 | Microglia in mouse aEAE and PEAE, and marmoset EAE, but not MS, increase TSPO expression. a** Representative images of TSPO+ microglia and astrocytes in MS. **b** TSPO+ microglia ($P = 0.2278$, $t = 1.306$, df = 8) and astrocytes ($P = 0.5476$, U = 9, ranks = 31, 24) do not increase TSPO expression in MS. **c** Representative images of TSPO expression in microglia and astrocytes in EAE mice. **d–f** microglia ($P < 0.0001$, $F_{(3,20)} = 25.68$), astrocyte ($P < 0.0001$, $F_{(3,20)} = 25.51$), and TSPO+ cell numbers ($P < 0.0001$, $F_{(3,20)} = 44.53$), are increased during disease in aEAE mice and PEAE. **g, h** An increase in both TSPO+ microglia ($P < 0.0001$, $F_{(3,20)} = 30.93$) and TSPO+ astrocytes ($P = 0.0005$, K-W = 17.72) is observed during disease. **i, j** TSPO+ microglia increase TSPO expression in aEAE mice ($P = 0.0136$, $t = 3.152$, df = 8), and in PEAE mice ($P = 0.0028$, $t = 4.248$, df = 8).

Astrocytes do not increase TSPO expression in aEAE ($P = 0.0556$, U = 3, ranks = 37, 18), and PEAE ($P = 0.5918$, $t = 0.5584$, df = 8). Biologically independent samples were used for all experiments (**b** $n = 5$ for all conditions) (**d–h** n = 6 for all groups) (**i, j** $n = 5$ CON and $n = 5$ aEAE/PEAE) Statistical significance in (**b**), (**d–h**) was determined by a one-way ANOVA or Kruskal–Wallis test when not normally distributed, and by a two-tailed unpaired $t$-test or two-tailed Mann–Whitney U-test when not normally distributed in (**i**) and (**j**). Holm–Sidak's and Dunn's multiple comparisons were performed. Box and whiskers mark the 25th to 75th percentiles and min to max values, respectively, with the median indicated. Scale bar = 50 μm, inserts are digitally zoomed in (200%). Each individual is represented by a different symbol. Source data are provided as a Source data file.

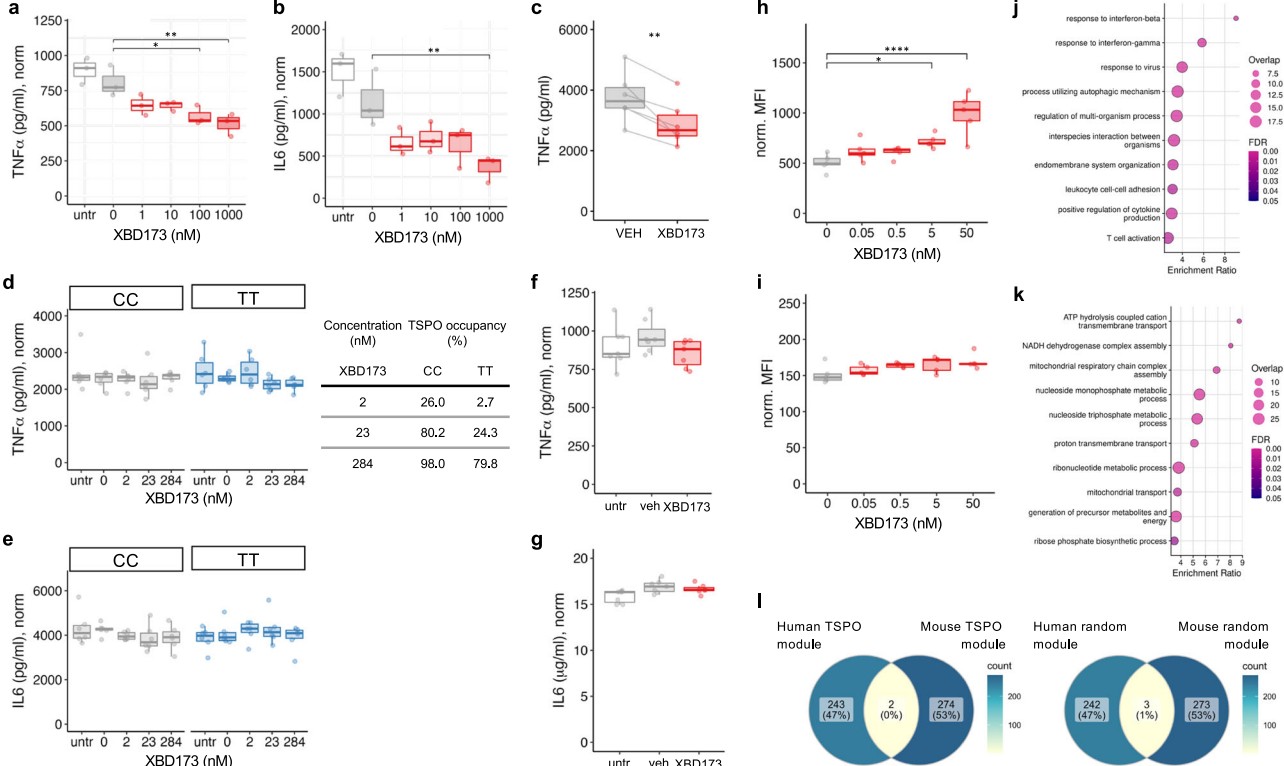

**Fig. 8 | TSPO ligand XBD-173 modulates classical pro-inflammatory myeloid cell function in mouse but not human myeloid cells. a–c** The specific TSPO ligand XBD-173 reduces LPS-induced cytokine secretion in mouse BV2 microglia (**a**, **b**) and primary bone marrow-derived macrophages (**c**; BMDM, XBD = 10 nM). (**a** $P = 0.0007$, F = 9.646, df = 5, padj$_{(100)}$ = 0.014, padj$_{(1000)}$ = 0.003; **b** $P = 0.0008$, F = 9.282, df = 5, padj$_{(1000)}$ = 0.006; **c** $P = 0.005$). **d–g** XBD-173 does not reduce LPS-induced cytokine secretion in human primary monocyte-derived macrophages from rs6971 C/C or T/T individuals (**d**, **e**) or in hiPSC derived microglia-like cells (**f**, **g**) C/C: $P = 0.833$, K-W = 1.46, df = 4; T/T: $P = 0.210$, K-W = 5.862, df = 4, $n = 6$; **e** C/C: $P = 0.10$, K-W = 7.780, df = 4, T/T: $P = 0.637$, K-W = 2.545, df = 4, $n = 6$; **f** $P = 0.057$, two-tailed $t$-test, XBD = 200 nM; **g** $P = 0.423$, XBD = 200 nM). **h, i** XBD-173 enhances phagocytosis in mouse BMDM (**h**) but not human monocytes (**i**) (**h** $P < 0.0001$, F = 12.07, df = 4; **i** $P = 0.173$, K-W = 6.38, df = 4). **j, k** TSPO gene co-expression

modules from naïve and pro-inflammatory primary macrophages in mouse and human. Gene ontology biological processes for the mouse TSPO module is enriched in classical pro-inflammatory pathways (**j**) and the human TSPO module is enriched for bioenergetic pathways (**k**). 2 genes overlap between mouse and human TSPO modules (**l**, left panel), compared to 3 genes overlapping between human and mouse random modules of the same size (**l**, right panel). Biologically independent samples were used for all experiments (**a**, **b** $n = 3$, **c–e** $n = 6$, **f**, **g** $n = 7$, **h**, **i** $n = 5$ for all conditions). Statistical significance in (**a**), (**b**), (**d**), (**e**), (**i**) and (**j**) was determined by one-way ANOVA or Kruskal–Wallis test when not normally distributed and by a two-tailed $t$-test in (**c**) (paired), (**f**) and (**g**) (both unpaired). Box and whiskers mark the 25th to 75th percentiles and min to max values, respectively, with the median indicated. Source data are provided as a Source data file.

were Reviewed and approved by the University of Edinburgh Animal Care Committee (ref.nr. ERF-096-KB-22). All animal procedures conformed to European Union directives and guidelines (EEC Council Directive, 63/2010/EU). Studies with TSPO-/- mice were carried out in accordance with the National Advisory Committee for Laboratory Animal Research guidelines and approved by the Nanyang Technological University, Singapore, Institutional Animal Care and Use Committee (IACUC) (ref.nr. #A0384).

Marmosets used for EAE studies were obtained in the United States and all marmosets were housed and handled with the approval of the NINDS/NIDCD/NCCIH Animal Care and Use Committee (ACUC).

All protocols were approved by the National Institutes of Neurological Disorders and Stroke (NINDS) Institutional Animal Care and Use Committee (IACUC) (ref.nr. #1308).

All human brain tissue was collected by brain banks or university medical centres and was approved by local and national guidelines. All participants or next of kin had given informed consent for autopsy and use of their tissue for research purposes. Netherlands Brain Bank (NBB): approved by Medical Ethics Committee (METC) Amsterdam UMC, Amsterdam, the Netherlands (ref.nr. 2009/148). London Neurodegenerative diseases brain bank: approved by Research Ethics Committee (REC), Wales REC 3 (ref.nr. 18/WA/0206). Queen

Square Brain Bank (QSBB): approved by London – Central REC, ref.nr. 08/H0718/54+5. Newcastle Brain Tissue Resource: approved by North East – Newcastle & North Tyneside 1 REC (ref.nr. 08/H0906/136+5). Manchester Brain Bank: approved by North East - Newcastle & North Tyneside 1 REC (ref.nr 09/H0906/52+5). Oxford Brain Bank: approved by South Central - Oxford C REC (ref.nr. 15/SC/0639). Multiple Sclerosis and Parkinson's Tissue Bank: approved by Wales REC 3 (ref.nr. 08/MRE09/31+5). The Geneva Brain collection: approved by the ethics commission of the Geneva Cantonal Commission of Ethics in Research (ref.nr. 2017-01598). Blood samples from healthy donors taken to isolate monocytes were drawn under a UK Research Ethics Committee approved protocol, Committee London – Bentham (ref.nr 12/LO/0538).

**Meta-analysis of TSPO gene expression.** Datasets were searched using the search terms "Macrophage/Monocyte/Microglia" and filtered for 'Homo sapiens' and 'Mus musculus'. Datasets with accessible raw data and at least three biological replicates per treatment group were used. To avoid microarray platform-based differences only datasets with Affymetrix chip were used. Raw microarray datasets were downloaded from ArrayExpress (https://www.ebi.ac.uk/arrayexpress/) and RMA normalisation was used. The 'Limma v.3.42.2' R package was used to compute differentially expressed genes, and the resulting P-values are adjusted for multiple testing with Benjamini and Hochberg's method to control the false discovery rate[70]. Meta-analysis was performed using R package 'meta v.5.1.1'. A meta P-value was calculated using the random-effect model.

**ChIP-seq data processing and visualization.** Raw fastq sequences for ChIP-seq datasets were aligned with Bowtie2 v.2.2.9[71] to the human reference genome hg19 or to mouse reference genome mm9, annotated SAM files are converted to tag directories using HOMER v.4.11.1[72] using the makeTagDirectory module. These directories are further used for peak calling using -style histone parameter or converted to the bigWig format normalized to $10^6$ total tag counts with HOMER using the makeUCSCfile module with -fsize parameter set at 2e9. For the analysis of histone ChIP-seq data input samples were utilized as control files during peak detection, whereas IgG control files were used during peak correction of the PU.1 ChIP-seq data. Peaks were visualised using UCSC genome browser[73].

**Multiple sequence alignment and phylogenetic tree construction.** We have retrieved the TSPO promoter region starting from 1000 bp upstream and 500 bp downstream of the putative transcription start site (TSS) of 34 rodent and non-rodent mammals from ENSEMBL genome database (http://www.ensembl.org/index.html). The full list can be found in Supplementary File 2. The multiple sequence alignment was performed using the T-Coffee (v13.45.0.4846264) multiple sequencing tool with the parameter -mode=procoffee which is specifically designed to align the promoter region[74,75]. The sequence alignment and the phylogenetic tree were visualised using Jalview (v 2.11.1.6)[76]. Phylogenetic tree was constructed using MEGA11 using Maximum Parsimony method with 1000 bootstrap replication. The MP tree was obtained using the Tree-Bisection-Regrafting (TBR) algorithm[77].

**Motif finding and motif enrichment.** We have used SEA (Simple Enrichment Analysis) from the MEME-suite (v5.4.1) to calculate the relative motif enrichment between Muroidea family species and non-Muroidea mammals[78,79]. We set the TSPO promoter sequences for the three Muroidea species (Mouse, Rat, Chinese Hamster) as the input sequence and the rest of species as the control sequence. We set the E-value ≤ 10 for calculating significance. We used the motifs for AP1, ETS (SPI1) and SP1 from JASPAR motif database (https://jaspar.genereg.net/).

**Multi-species TSPO expression in macrophage and microglia.** Datasets were searched using the search terms "Macrophage/Monocyte", "Microglia" and "LPS". Dataset featuring stimulation less than 3 h were excluded. Datasets with accessible raw data and at least three biological replicates were used. Microarray datasets were analysed as the same way described in section "Meta-analysis of TSPO gene expression". Raw gene count data for the RNAseq datasets were downloaded from either ArrayExpress or GEO (https://www.ncbi.nlm.nih.gov/geo/) and differential expression was performed using DESeq2 v.1.26.0[80]. For S3a, the mouse Tspo expression (https://www.ncbi.nlm.nih.gov/geo/query/acc.cgi?acc=GSE38371) fold change was directly used from the respective study since biological replicates were not publicly accessible[30].

**Identification of genes with an AP1 binding site.** AP1 dimeric transcription factor is composed of Jun and Fos protein families, so we retrieved the Jun (JUN, JUNB, JUND) and Fos (FOS, FOSL1, FOSL2) binding regions of the human genome GRCh38 assembly as defined by the ENCODE3 TF Clusters (https://genome.ucsc.edu/cgi-bin/hgTrackUi?db=hg38&g=encRegTfbsClustered) from UCSC genome browser (https://genome.ucsc.edu/) and filtered for the genes which have a binding site within the 2 kb upstream and 1 kb downstream of TSS (transcription start site).

**Human and mouse scRNAseq and snRNAseq analysis of microglia**
Raw count matrices were processed with Seurat (v3)[81] or nf-core/scflow[82]. Quality control, sample integration, dimension reduction and clustering were performed using default parameters as previously described[41,83]. The human snRNAseq dataset (https://www.ncbi.nlm.nih.gov/geo/query/acc.cgi?acc=GSE160936) was processed as previously described[41]. For the human scRNAseq dataset[39] (https://www.synapse.org/#!Synapse:syn21438358), a k integration parameter of 40 was employed, otherwise the processing was performed with default parameters. The mouse scRNAseq dataset (https://www.ncbi.nlm.nih.gov/geo/query/acc.cgi?acc=GSE98969) was processed using scFlow using default parameters. Microglial cells/nuclei were identified using previously described cell markers. To distinguish between homeostatic and pro-inflammatory/disease-associated microglial subclusters, we calculated the enrichment of cells/nuclei in homeostatic, pro-inflammatory[84] and disease-associated[40] microglial markers (using AUCell[85]) and performed pairwise comparisons of their enrichment between subclusters (using limma[86]). We then performed differential gene expression analysis between the pro-inflammatory/disease-associated and the homeostatic microglia using MAST[87] implemented in scFlow by fitting a mixed-effects zero-inflated negative binomial model with the cluster and the total number of features expressed per cell/nucleus as a fixed-effects variables and the individual sample as a random-effects variable. In all cases, we confirmed that individual markers of activated microglia were upregulated in the pro-inflammatory/disease-associated microglia and homeostatic markers were downregulated. For the human snRNAseq datasets, the statistical analyses (i.e. the pro-inflammatory/disease-associated vs homeostatic and the BD vs control comparisons) were performed in pseudobulked expression matrices, given the low detection of TSPO in single nuclei. Pseudobulking bioinformatically transforms snRNAseq data to single cell type "bulk" RNAseq data by summing expression values over multiple nuclei per sample, and performs the analyses on individual samples rather than on individual nuclei. This pseudobulking approach is integrated in the MAST implementation in scFlow. For the bipolar disorder analysis, 12 samples of cingulate gyrus were obtained from the Netherlands Brain Bank in Amsterdam from individuals with bipolar disorder (type I or type II, $n = 7$) and from non-neurological, non-psychiatric controls ($n = 5$). Demographic details are listed in

Table S11. The same protocol for the negative enrichment of nuclei using FACS prior to barcoding and sequencing as detailed in Smith et al.[41] was employed. Data analysis and quantification was performed as described for above in pseudobulked expression matrices. Gene expression alterations were considered significant when the adjusted $p$ value was equal to or lower than 0.05.

**Brain cell type expression specificity assessment for HK2, LCP2 and TFEC.** To assess the cell specificity of expression of HK2, LCP2 and TFEC in the CNS we used a previously described human AD snRNAseq dataset[88]. We used MAST to perform pairwise comparisons of the expression of *HK2, LCP2* and *TFEC* in microglia vs each one of the other cell types.

**Bulk RNA-seq data preparation and WGCNA network analysis.** RAW RNA-seq fastq files for publicly available datasets were downloaded from SRA. Four public human dataset accession are: https://www.ncbi.nlm.nih.gov/geo/query/acc.cgi?acc=GSE100382, https://www.ncbi.nlm.nih.gov/geo/query/acc.cgi?acc=GSE55536, https://www.ebi.ac.uk/biostudies/arrayexpress/studies/E-MTAB-7572?query=E-MTAB-7572, https://www.ncbi.nlm.nih.gov/geo/query/acc.cgi?acc=GSE57494 and one in-house dataset https://www.ncbi.nlm.nih.gov/geo/query/acc.cgi?acc=GSE236998. Mouse dataset accession are: https://www.ncbi.nlm.nih.gov/geo/query/acc.cgi?acc=GSE103958, https://www.ncbi.nlm.nih.gov/geo/query/acc.cgi?acc=GSE62641, https://www.ncbi.nlm.nih.gov/geo/query/acc.cgi?acc=GSE82043, https://www.ncbi.nlm.nih.gov/geo/query/acc.cgi?acc=GSE58318, https://www.ebi.ac.uk/biostudies/arrayexpress/studies/E-ERAD-165?query=E-ERAD-165%20. Both human and mouse RNA-seq analysis was then performed using nf-core/rnaseq v.1.4.2 pipeline[89]. Human RNA-seq data was aligned to Homo sapiens genome GRCh38 and Mus musculus genome mm10, respectively. Raw count data was first transformed using variance stabilizing transformation (VST) from R package 'DESeq2 v. 1.26.0'. Genes with an expression value of 1 count in at least 50% of the samples were included in the analysis. Batch correction across datasets were then performed on VST-transformed data using removeBatchEffect function from R package 'Limma v. 3.42.2' using the dataset ID as the batch. Batch-corrected normalised data was then used for co-expression network analysis using the R package 'WGCNA v. 1.69'[90]. The power parameter ranging from 1–20 was screened out using the 'pickSoftThreshold' function. A suitable soft threshold of 6 was selected, as it met the degree of independence of 0.85 with the minimum power value. We generated a signed-hybrid network using Pearson correlation with a minimum module size of 30. Subsequently, modules were constructed, and following dynamic branch cutting with a cut height of 0.95. Functional enrichment analysis of the gene modules was performed using the R package 'WebGestaltR v. 0.4.3'[91] using default parameters and 'genome_protein-coding' as the background geneset.

**Human brain tissue**
For the samples used in the ALS cohort: Human tissue was obtained at autopsy at the Department of Neuropathology of the Amsterdam UMC (University of Amsterdam, the Netherlands). In this study, we included tissue from the spinal cord (cervical, thoracic, lumbar levels) from 12 ALS patients, 7 with short disease duration (SDD; <18 months survival; mean survival 11.1 ± 3.4 months) and 4 with medium disease duration (MDD; >24 months survival; mean survival 71.5 ± 31.5 months). Tissues for controls were collected from 10 age-matched cases with no neurological disorders or peripheral inflammation (Table S5).

For the samples used in the IF only cohorts (AD and MS): The rapid autopsy regimen of the Netherlands Brain Bank in Amsterdam (coordinator Prof I. Huitinga) was used to acquire the samples at the Department of Neuropathology of the Amsterdam UMC (University of Amsterdam, the Netherlands). The hippocampal region (CA4) was collected from 5 AD patients with Braak stage 6, and 5 aged-matched controls that had no cognitive impairments prior to death (Table S1). Active MS lesions were obtained from 5 MS cases as well as white matter from age-matched controls (Table S6). For the AD samples used in the confocal and autoradiography cohort: The human brain samples were obtained from the Geneva Brain Bank. A board-certified neuropathologist determined the Braak stage for neurofibrillary tangles. Demographic details are listed in Tables S2 and S3.

For the AD samples used in the IMC cohort: Cases were selected based first on clinical and neuropathological diagnosis from UK brain banks (London Neurodegeneration [King's College London], Newcastle Brain Tissue Resource, Queen's Square Brain Bank [UCL], Manchester Brain Bank, Oxford Brain Bank and Parkinson's UK [Imperial College London] Brain Bank). We then excluded cases with clinical or pathological evidence for small vessel disease, stroke, cerebral amyloid angiopathy, diabetes, Lewy body pathology (TDP-43), or other neurological diseases. The region of interest was the mid-temporal gyrus. Demographic details are listed in Table S4.

**Generation and details of mouse and marmoset models**
**Mouse EAE.** Spinal cord tissue from mice with EAE was obtained from Biozzi ABH mice housed at Queen Mary University of London, UK (originally obtained from Harlan UK Ltd, Bicester, UK). The mice were raised under pathogen-free conditions and showed a uniform health status throughout the studies. EAE was induced via injection of mouse spinal cord homogenate in complete Freund's adjuvant (CFA) into mice of 8–12 weeks or 12 months of age as described previously[45,92]. Immediately, and 24 h after injection mice were given 200 ng *Bordetella pertussis* toxin (PT). Age-matched control groups were immunized with CFA and PT. Table S6 gives an overview of the EAE mice used in this study, including a score of neurological signs (0 = normal, 1 = flaccid tail, 2 = impaired righting reflex, 3 = partial hindlimb paresis, 4 = complete hindlimb paresis, 5 = moribund). Spinal cord was collected from acute (aEAE)[92] in the young mice, and progressive EAE (PEAE) in the 12-month-old mice[93].

**Marmoset EAE.** EAE was induced by subcutaneous immunization with 0.2 g of white matter homogenate emulsified in CFA in 3 adult common marmosets (*Callithrix jacchus*) at 4 dorsal sites adjacent to inguinal and axillary lymph nodes. Animals were monitored daily for clinical symptoms of EAE progression and assigned clinical EAE scores weekly based on extent of disability. Neurological exams were performed by a neurologist prior to each MRI scan. All animals discussed in this study are shown in Table S7. Animal #8 was treated with prednisolone for 5 days as part of a concurrent study (primary results not yet published). These animals were the first within their twin pair that showed three or more brain lesions by in vivo MRI and received corticosteroid treatment with the goal to reduce the severity of inflammation and potentially allow longer-term evaluation of the lesions. MRI analyses were performed according to previously published marmoset imaging protocols using T1, T2, T2*, and PD-weighted sequences on a Bruker 7T animal magnet[94]. Marmosets were scanned biweekly over the course of the EAE study. Following the completion of EAE studies, the brains, spinal cords, and optic nerves excised from euthanized animals were scanned by MRI for *postmortem* characterization of brain lesions and previously uncharacterized spinal lesions and optic nerve lesions.

**SOD1**^G93A. Female hemizygous transgenic SOD1^G93A mice on 129SvHsd genetic background ($n = 10$) and corresponding non-transgenic littermates ($n = 9$) were used. This mouse line was raised at the Mario Negri Institute for Pharmacological Research-IRCCS, Milan, Italy, derived from the line (B6SJL-TgSOD1^G93A-1Gur, originally purchased from Jackson Laboratories, USA) and maintained on a 129S2/SvHsd background[95]. The thoracic segments of spinal cord were collected

from 10- and 16-week-old mice and processed as previously described[96]. Briefly, anaesthetised mice were transcardially perfused with 0.1 M PBS followed by 4% PFA. The spinal cord was quickly dissected out and left PFA overnight at 4 °C, rinsed, and stored 24 h in 10% sucrose with 0.1% sodium azide in 0.1 M PBS at 4 °C for cryoprotection, before mounting in optimal cutting temperature compound (OCT) and stored at −80 °C.

**$APP^{NL-G-F}$.** For the $APP^{NL-G-F}$ model of AD, male and female brain tissue was obtained from 11 homozygous ($APP^{NLGF/NL-G-F}$) APP knock-in mice and 11 wild-type mice. Mice were bred at Charles River Laboratories, UK and sampled at the Imperial College London, UK. Brain tissue samples were collected fresh from 10- and 28-week-old mice that were euthanised with sodium pentobarbital and exsanguinated. Hippocampal areas were used as region of interest for characterization.

**$Tau^{P301S}$.** Male brain tissue was obtained from 10 homozygous P301S knock-in mice[97–99] and 8 wild-type C57/Bl6-OLA mice (Envigo, UK) from the Centre for Clinical Brain Sciences, Edinburgh, United Kingdom. Brain tissue samples were collected from 8- and 20-week-old mice that were perfused with PBS and 4% paraformaldehyde, with tissues being post-fixed overnight before being cryopreserved in 30% sucrose and frozen embedded in tissue tec (Leica, UK). Sections were cut, 20 μm, on a cryostat onto superfrost plus slides and stored in −80 freezer. Hippocampal areas were used as region of interest for characterization.

**$TSPO^{-/-}$ mice.** Male and female adult TSPO-KO mice[100] (>6 months age) and C57BL6 mice were used to test the specificity of the TSPO antibody. At the day of sacrifice, mice were anesthetized with sodium pentobarbital (200 mg/kg i.p.) and were either transcardially perfused or decapitated followed by immersion fixation in 4% paraformaldehyde for 24 h for cryopreservation. After perfusion, the brains were extracted and post-fixed in 4% paraformaldehyde for 24 h. All mice were treated with LPS (1 mg/kg, IP) for 3 consecutive days to increase TSPO expression.

For all studies mice were housed 4–5 per standard cages in specific pathogen-free and controlled environmental conditions (temperature: 22 ± 2 °C; relative humidity: 55 ± 10% and 12 h of light/dark). Food (standard pellets) and water were supplied ad libitum. A detailed overview of tissue collection and storage is displayed in Fig. S19. Methods Models.

**Immunohistochemistry and immunofluorescence.** Immunofluorescence was performed on serial sections for all staining procedures. Paraffin sections (5 μm) were de-paraffinized by immersion in xylene for 5 min and rehydrated in descending concentrations of ethanol and fixed-frozen sections were dried overnight. After washing in PBS, sections were incubated in 0.1% glycine. Antigen retrieval was performed with citrate or TRIS/EDTA buffer, depending on the antibody, in a microwave for 3 min at 1000 W and 10 min at 180 W. Sections were cooled down to RT and incubated with primary antibodies (Table S9) diluted in antibody diluent (Sigma, U3510) overnight. Sections were washed with PBS and afterwards incubated with the appropriate fluorescent secondary antibodies for 1 h at room temperature. Autofluorescent background signal was reduced by incubating sections in Sudan black (0.1% in 70% EtOH) for 10 min. Nuclei were stained with 4,6-diami-dino-2-phenylindole (DAPI) and slides were mounted onto glass coverslips with FluoromountTM (Merck). For the LCP2 experiment in AD tissue, HRP labelled antibodies were developed with diluted 3,3'-diaminobenzidine (DAB; 1:50, DAKO) for 10 min and counterstained with haematoxylin. Sections were immersed in ascending ethanol solutions and xylene for dehydration and mounted with Quick-D.

**Confocal microscopy.** FFPE slices were deparaffined and pretreated with 0.1% glycine in $H_2O$ for 10 min at room temperature (RT). Following a 70% formic acid treatment (4 min, RT), slices were rinsed in $H_2O$. Slices were immersed in citrate buffer (0.01 M, pH = 6) and placed in a cooker for 20 min at 95 °C. Slices were rinsed in $H_2O$ and treated overnight at 4 °C in 0.1 M PBS-1% BSA-0.3% Triton X-100 with the following antibodies: goat anti-IBA1 (Ab48004, Abcam, 1/300) and rabbit anti-TSPO (Ab109497, Abcam, 1/300). Slices were rinsed in 0.1 M PBS (3 × 10 min) and treated with the secondary antibodies (donkey anti-goat 488 and donkey anti-rabbit 555, Thermofisher, 1/300) in 0.1 M PBS-1% BSA-0.3% Triton X-100 for 90 min at RT. After rinsing in 0.1 M PBS (3 × 10 min), slices were mounted in Fluor save. The Axio Imager.Z2 Basis LSM 800 microscope (Zeiss) was used to take pictures at CA4 (stack of images every 0.3 μm). A pixel-based analysis in the different channels on each Z-position was realized to calculate the % of colocalization (average of average of three distinct measures per individual) using imageJ (v1.53c). The representative image shows the maximum intensity projection (Z-stack Processing, ImageJ).

**Autoradiography.** To estimate the densities of TSPO binding sites, in situ autoradiography was performed as previously described using [$^{125}$I]CLINDE[101]. Diluted reaction with 50% acetonitrile (ACN) was purified using a linear gradient HPLC run (5% to 95% ACN in 7 mM H3PO4, 10 min) with a reversed-phase column (Bondclone C18). Brain samples were incubated with 0.05 MBq/ml of radiotracer with specific activity greater than 650 GBq/μmol, based on the limit of detection of the ultraviolet absorbance and on the calibration curves established with cold reference compounds. Serial coronal brain sections (20 μm) were cut on a cryostat and slices were incubated in a Tris buffer (50 mM Tris HCl, pH 7.4) alone (20 min), then in the same buffer containing [$^{125}$I]CLINDE (90 min) then rinsed twice in 4 °C buffer (3 min) and briefly washed in cold water. Non-specific binding was estimated in the presence of 10 μM of unlabelled ligands on adjacent sections. Air-dried slides were then exposed to gamma-sensitive phosphor imaging plates (Fuji BAS-IP MS2325) and resulting autoradiograms were analysed with Aida Software V4.06 (Raytest Isotopenmessgerate GmbH) together with homemade 125I calibration curves. The specific binding ratio (SBR) was calculated as follows: (hippocampus/hippocampus with 10 μM of unlabelled radiotracer). Statistical analysis was performed by fitting a linear model using the Braak stage as the comparator and the TSPO as a confounder and performing a two-way ANOVA.

**Imaging mass cytometry (human).** Antibody conjugation was performed using the Maxpar X8 protocol (Fluidgm). Slides with paraffin-embedded tissue from the Medial Temporal Gyrus (MTG) of 22 AD donors underwent IMC staining and ablation. Each slide was within 5–10 μm in thickness. The slides underwent routine dewaxing and rehydration before undergoing antigen retrieval in a steamer (100 °C) for 20 min, in a pH 8 Ethylenediaminetetraacetic acid (EDTA) buffer. The slides were blocked in 10% normal horse serum (Vector Laboratories) before incubation with a conjugated-antibody cocktail (Table S8) at 4 °C overnight. Slides were then treated in 0.02% Triton X-100 (Sigma-Aldrich) before incubation with an Iridium-intercalator (Fluidgm, 1:400) then washed in dH₂O and air-dried overnight. Image acquisition took place using a Hyperion Tissue Imager (Fluidgm) coupled to a Helios mass cytometer. The instrument was tuned using the manufacturer's 3-Element Full Coverage Tuning Slide before the slides were loaded into the device. Four 500 × 500 μm regions of interest within the grey matter were selected and then ablated using a laser at a frequency of 200 Hz at a 1 μm resolution. The data was stored as .mcd files compatible with MCD Viewer software (Fluidgm) then exported as TIFF files. Post-acquisition image processing using ImageJ (v1.53c) software allowed threshold correction and the despeckle function to reduce background noise. The data was opened with

HistoCAT (BodenmillerGroup) to quantify the signal of each Ln-channel and exported as .csv files.

**Pan-cellular masking.** The IMC channels which define a cell type were selected for masking, namely IBA1, GFAP, MAP2, GLUT1, OLIG2 and p16. All six of these channels were merged into a single field, alongside a DNA channel, on ImageJ (v1.53c). The DNA channel was rendered red, while the cell-type channels were rendered green before being saved as a JPEG. This JPEG was then used in Ilastik (v1.1.3post3), a probability map was created with the pixel classification tool, clearly defining the images nuclei, cell signal and background. Finally, this probability map was imported to CellProfiler (v4.2.1) and was masked by using a Cell-Profiler pipeline based on a foundation of a pipeline created previously by the Bodenmiller group[102]. Once masked the sample was opened in HistoCAT (v1.76) and the masked cell data was exported as a csv, with quantitative values for the signal of each IMC channel for each cell.

**Microglia masking.** The IMC channels IBA1 and DNA were merged into a single field, on ImageJ (v1.53c). The DNA channel was rendered red, while the IBA1 channel was rendered green before being saved as a JPEG. This JPEG was then used in Ilastik (v1.1.3post3), a probability map was created using the pixel classification tool, clearly defining the images nuclei, cell signal and background. Finally, this probability map was imported to CellProfiler (v4.2.1) and was masked by using a Cell-Profiler pipeline based on a foundation of a pipeline created previously by the Bodenmiller group. The masking process first identified the nuclei as 'primary objects', then expanded outwards from the nuclei into the surrounding cell to until it reached the cell membrane to define the 'secondary object'. By using the nuclei as primary objects the risk of identifying background or artefact as cell bodies was negated. Once masked the sample was opened in HistoCAT (v1.76) and the masked cell data was exported as a csv, with quantitative values for the signal of each IMC channel for each cell.

**Multiplex immunofluorescence (marmoset).** To immunophenotype microglia and macrophages expressing TSPO in the marmoset CNS, a multi-colour multiplex immunofluorescence panel was used to stain for IBA1, PLP, and TSPO. Deparaffinised sections were washed twice in PBS supplemented with 1 mg/ml BSA (PBS/BSA), followed by two washes in distilled water. Antigen retrieval was performed by boiling the slide in 10 mM citrate buffer (pH 6) for 10 min in an 800 W microwave at maximum power, after which they were allowed to cool for 30 min and washed twice in distilled water. To reduce non-specific Fc receptor binding, the section was incubated in 250 µl of FcR blocker (Innovex Biosciences, NB309) for 15 min at room temperature and washed twice in distilled water. To further reduce background, sections were coated with 250 µl Background Buster (Innovex Biosciences, NB306) for 15 min at room temperature and washed twice in distilled water. Sections were incubated for 45 min at room temperature in a primary antibody cocktail containing antibodies diluted in PBS/BSA (Supplemental Table 1), washed in PBS/BSA and three changes of distilled water. They were then incubated for 45 min in a secondary antibody cocktail composed of secondary antibodies diluted in PBS/BSA containing DAPI (Invitrogen, cat. no. D1306, 100 ng/mL) (Supplemental Table 2), then washed once in PBS/BSA and twice in distilled water. To facilitate mounting, the sections were air-dried for 15 min at room temperature, sealed with a coverslip as described previously, and allowed to dry overnight prior to image acquisition.

**Immunohistochemistry and Immunofluorescence imaging and statistical analyses.** Brightfield images were collected at ×40 magnification using a Leica DC500 microscope (Leica Microsystems, Heidelberg, Germany, Japan), or a Leica DM6000 (Leica Microsystems, Heidelberg, Germany) or a Zeiss AxioImager.Z2 wide field scanning microscope for fluorescent images (gamma 1.0, binning 1 × 1, HCX PL APO 40.0 × 0.85 Objective, RI 1.0). No shading correction, brightness correction, or artificial gain was used for images. For AD, APP[NL-G-F], and TAU[P301S] tissue images were collected from the CA4 region of the hippocampus. For ALS tissue, images of the ventral horn and the lateral column were obtained from cervical, thoracic, and lumbar spinal cord levels. For mouse EAE and SOD1[G93A] mice, images of grey and white matter of the spinal cord were collected per case. In all instances, 5 random images (300 × 250 µm) within the region of interest per disease or animal model were made after which ImageJ software was used for picture analyses. Nuclei and stained cells were counted manually using the cell counter plugin (de Vos, University of Sheffield, UK), excluding nuclei at the rim of each picture and within blood vessels and nuclei which overlap. To determine inter-observer variation 18 pictures were manually counted by 3 independent observers with a correlation coefficient of >0.9. To determine single-cell TSPO expression, IBA1+ or GFAP+ cells were outlined manually using the imageJ ROI manager (Fig. S20). Afterwards, TSPO+ pixels were measured within IBA1+ and GFAP+ ROIs per cell. Raters were blinded to disease status. Data were analysed using GraphPad Prism 9.1.0 software. All data were tested for normal distribution, using the Shapiro-Wilk normality test. Significant differences were detected using an unpaired $t$-test or one-way analysis of variance test. Dunnett's post-hoc test was performed to analyse which groups differ significantly. Number of mice was calculated by power analysis and as a maximum of 6–8 mice were used per group based on previous studies[45]. Data was considered significant when $P < 0.05$.

For the quantification of TSPO near amyloid plaques and NFT, quadruple immunofluorescence staining was performed for TSPO, amyloid plaques (using MX04), pTau and IBA1 in a cohort of hippocampus samples from AD patients and NDC. Analysis was performed in ImageJ. After thresholding (a common value was employed for all the images for each marker), ROI 10 µm around individual plaques and NFT that were bordering a random grid (10 per individual samples). For each plaque, an ROI of at least 50 µm far from any lesion was also delineated. The % of the total IBA1 positive area that colocalized with TSPO was measured in the ROI around the lesions and the ROI far from them and the results were computed using a mixed-effects model with the distance from the lesions as a main comparator and the individual sample as a random-effect variable.

For the examination of LCP2 in AD, the whole hippocampus was analysed by manually delineated cyclical ROI. A threshold for LCP positivity was estimated on each sample and then the average values across all samples was applied on each image. The % LCP2 positive area was then measured, and the results were compared between AD and NDC samples using a one-way ANOVA.

Given that double immunofluorescence was performed in the MS and ALS cohorts, LCP2 staining per cell and total microglial LCP2 staining were estimated. For the analysis of double IBA1 and LCP2 immunofluorescence in MS, IBA1-positive cell bodies bordering a random grid were selected. The optical density in these IBA1 positive areas was measured in binary converted images in the LCP2 channel. A total of 195 IBA1-positive cells were evaluated in this manner. At least 3 images were acquired per patient and per lesion (or NAWM, respectively) and finally the mean values of the patients were statistically analysed using a one-way ANOVA. In the ALS cohort, to measure LCP2 binding per microglial cell, cellular LCP2 was estimated in QuPath (v0.4.2) after manually training a cell classifier that identified IBA1-positive cells. Given that the LCP2 staining was nuclear, LCP2 staining was measured in DAPI stained nuclei associated to IBA1 positive cells and the results were compared between ALS and NDC samples using a mixed-effects model with diagnosis as main comparator, region (ventral horn and lateral columns) as confounder and the samples as random-effect variable.

To estimate total microglial LCP2 binding in MS and ALS, a pixel classifier was trained in QuPath (v4.2.0) and then the percentage of the total area (lesions or NAWM for the MS and ventral horn and lateral

column for the ALS cohort) covered by double LCP2 and IBA1 positive pixels was estimated. Statistical comparisons were performed as described above.

**IMC statistical analysis.** The values corresponding to the density of TSPO, HLA-DR, CD68, beta amyloid and pTau from each cell were analysed using a zero-inflated mixed-effects gamma distribution model using the glmmTMB R package (10.32614/RJ-2017-066) and a type II Anova test (Anova function from the car R package), where the proximity to amyloid plaques and NFT was used as fixed effects variable and the individual donor was the random-effect variable. The same model was also employed including the TSPO genotype as fixed effects variable. Pearson's correlation coefficients between TSPO, CD68 and HLA-DR were calculated from the individual cell IMC data.

**BV2 and primary mouse macrophage and microglia culture.** Cells were kept at 37 °C, 5% $CO_2$ and 95% humidity. Mouse BV2 cells (a kind gift from Federico Roncaroli, Manchester) were cultured in RPMI-1640 containing 2 mM GlutaMAX and 10% heat inactivated FBS (all Gibco). For experiments, BV2 were seeded at $1 \times 10^4$ cells per well of a 96-well plate the day before treatment. Primary mouse bone marrow-derived macrophages (BMDMs) were obtained from bone marrow of adult C57BL/6 mice and cultured in DMEM containing 10% FBS, penicillin/streptomycin, and glutamine supplemented with M-CSF (10 ng/mL; Peprotech) as previously described[103]. For microglia, mixed glia cultures were derived from P0-P5 C57Bl/6 mouse pups purchased from Charles River and cultured in DMEM supplemented with 10% FBS, 100 U/mL penicillin and 100 µg/mL streptomycin and 2 mM glutamine (all from Invitrogen). At confluency, cultures were subjected to mild trypsinization. Following removal of the astrocyte monolayer, microglia were re-plated at a density of $2 \times 10^5$ cells/mL in DMEM + 10% FBS, 100 U/mL penicillin and 100 µg/mL streptomycin and 2 mM glutamine (all from Invitrogen) with a 1:1 ratio with astrocyte conditioned media (media collected from confluent rodent astrocytes) prior to stimulation. Microglia were allowed to adhere for at least 24 h before prior to experimentation. There was no gender bias. Red blood cells were lysed in an ammonium chloride solution, and the remaining bone marrow cells were cultured at $2.5 \times 10^6$ cells/mL in DMEM supplemented with 10% FBS, 100 U/mL penicillin and 100 µg/mL streptomycin and 2 mM glutamine (all from Invitrogen) and M-CSF (10 ng/mL). All animal procedures were approved by the Memorial University Animal Care Committee in accordance with the guidelines set by the Canadian Council in Animal Care.

**Primary human macrophage and microglia culture.** All donors gave informed consent under a REC approved protocol (12/LO/0538). Human monocyte-derived macrophages (MDMs) were obtained from fresh blood of male and female, healthy donors between 20 and 60 years after CD14-affinity purification. In brief, whole blood was diluted 1:1 with DPBS (Sigma), layered onto Ficoll (Sigma) and spun for 20 min at $800 \times g$ with minimal acceleration/deceleration. Peripheral mononuclear cells were collected, washed, and labelled with CD14-affinity beads (Miltenyi) according to the manufacturers protocol. CD14 monocytes were eluted and cultured at $5 \times 10^5$ cells/ml in RPMI-1640 containing 2 mM GlutaMAX, 10% heat inactivated FBS, and 25 ng/mL M-CSF (all Gibco) with medium change after 3 days. MDMs were used after 7 days in vitro culture. For monocytes, M-CSF was omitted from the medium and cells were used immediately ex vivo. Adult microglia were isolated from a mixture of white and grey matter of temporal lobe brain tissue, from patients undergoing surgery for intractable epilepsy not related to tumours in accordance with the guidelines set by the Biomedical Ethics Unit of McGill University (ANTJ2001/1). All experiments were conducted in accordance with the Helsinki Declaration. Written, informed consent was obtained from all subjects. There was no gender bias. The tissue provided was outside of the suspected focal

site of epilepsy-related pathology. The tissue was processed as previously described[104]. Briefly, tissue was obtained in pieces <1 mm³ and treated with DNase (Roche, Nutley, NJ) and trypsin (Invitrogen, Carlsbad, CA) for 30 min at 37 °C. Following dissociation through a nylon mesh (100 µm), the cell suspension was separated on a 30% Percoll gradient (GE Healthcare, Piscataway, NJ) at $31,000 \times g$ for 30 min. Glial cells (oligodendrocytes and microglia) were collected from underneath the myelin layer, washed, and then plated in tissue-culture-treated vessels. Floating oligodendrocytes were washed off on the subsequent day and the remaining adherent microglia were collected with trypsin and 2 mM EDTA (Sigma-Aldrich). Cells were plated at $1 \times 10^5$ cells/mL in minimum essential medium (Sigma) containing 5% foetal bovine serum (FBS), 100 U/mL penicillin and 100 µg/mL streptomycin and 2 mM glutamine (all from Invitrogen). The proportion of microglia in the culture, determined using CD11c staining by flow cytometry, was ≥90%[104].

**Human TSPO genotyping.** Genotyping at rs6971 was performed by LGC. Where not specified, studies were performed with homozygous A carriers due to the high affinity for XBD-173.

**iPSC culture and microglia-like cell differentiation.** The human induced pluripotent stem cell (iPSC) line SFC841-03-01 (https://hpscreg.eu/cell-line/STBCi044-A, previously derived from a healthy donor[105], Oxford Parkinson's Disease Centre/StemBANCC) was obtained under MTA from the James Martin Stem Cell Facility, University of Oxford and cultured in feeder-free, fully defined conditions. In brief, iPSCs were maintained in E8 medium on Geltrex (both Gibco) and fed every day until 80% confluent. For cell cluster propagation, iPSCs were lifted with 0.5 mM EDTA (Thermo) in DPBS and upon visible dissociation, EDTA was removed, and iPSC were diluted 4–6 times in E8 for culture maintenance. iPSCs were screened genotypically for chromosomal abnormalities using single nucleotide polymorphism analysis and phenotypically using Nanog (Cell Signalling) and Tra-1-60 (BioLegend) immune positivity. Mycoplasma infection was excluded based on LookOut test (Sigma) according to manufacturer's protocol. Microglia-like cells were differentiated according to Haenseler et al.[106]. In short, on day 0 iPSCs were dissociated with TrypLE Express (Gibco) and $4 \times 10^6$ iPSCs were added to one well of 24-well AggreWell™ 800 (Stem Cell Technology) according to the manufacturer's protocol in 2 ml EB medium (E8, SCF (20 ng/mL, Miltenyi), BMP4 (50 ng/ml; Gibco), VEGF (50 ng/mL, PeproTech)) with 10 µM ROCK inhibitor (Y-27632, Abcam). From day 1 to 6, 75% medium was exchanged with fresh EB. On day 7 embryoid bodies were transferred to $2 \times$ T175 flasks containing factory medium (XVIVO-15 (Lonza), 2 mM GlutaMAX, 50 µM 2-Mercaptoethanol, 25 ng/mL IL-3, and 100 ng/mL M-CSF (all Gibco)) and fed weekly with factory medium. Starting from week 4 after transfer, medium was removed and tested for the presence of primitive macrophages using CD45 (immunotools), CD14 (immunotools) and CD11b (Biolegend) immunopositivity by flow cytometry (FACSCalibur, BD Biosciences). Primitive macrophages were transferred to microglia medium (SILAC Adv DMEM/F12 (Gibco), 10 mM glucose (Sigma), 2 mM GlutaMAX, 0.5 mM L-lysine (Sigma), 0.5 mM L-arginine (Sigma), 0.00075% phenol red (Sigma), 100 ng/mL IL-34 (PeproTech), 10 ng/mL GM-CSF (Gibco)), fed every 3–4 days and used for experiments after 7 days.

**Drug treatments and cell activation.** Cells were treated with XBD-173 at the indicated concentrations for 1 h prior to LPS activation or for 20 h prior to phagocytosis. Pro-inflammatory activation was induced with lipopolysaccharide (10 ng/mL or 100 ng/mL; Sigma) IFNγ or IFNα (both at 1 ng/mL and 10 ng/mL; Sigma) for 4 h, 8 h or 24 h as indicated. For live-cell phagocytosis assays, pHrodo®-labelled zymosan A bioparticles (Thermo) were added to the culture medium and incubated for 2 h at 37 °C with 5% $CO_2$. pHrodo®-fluorescence intensity was

acquired in a plate reader (Cytation5, BioTek) or by Flow cytometry (FACSCalibur, BD Biosciences).

**Cytokine analysis.** Cytokines were assessed from cell-free cell culture supernatant using enzyme-linked immunosorbent assay (ELISA) according to the manufacturers' protocols. The following assays were used: mouse-TNFα and mouse-IL-6 ELISA (R&D Systems), human-TNFα and human-IL-6 (BD Biosciences). Absorbance was measured in a Spark plate reader (Tecan).

**Gene expression.** 200 ng of RNA was reverse-transcribed using M-MLV reverse transcriptase (ThermoFisher) and random hexamers. qPCR reactions were carried out using TaqMan universal master mix and gene specific primers. mRNA expression was normalized to gapdh (Hs02786624_g1 and Mm99999915_g1). Fold changes were calculated using the delta delta CT method. The gene specific primers used were TSPO (Hs00559362_m1, Mm00437828_m1), Ifi44 (Hs00197427_m1, Mm00505670_m1), TFEC (Hs00992838_m1, Mm01161234_m1), HK2 (Hs00606086_m1, Mm00443385_m1), and LCP2 (Hs01092638_m1, Mm01187570_m1).

**RNA Sequencing.** RNA was extracted from control and LPS treated (100 ng/mL, 24 h) primary human macrophages using the RNeasy Mini Kit. cDNA libraries (Total RNA with rRNA depletion) were prepared and sequenced using a HiSeq4000. Lanes were run as 75 bases Paired End. Sequencing depth was minimum 40 million reads per sample.

**Western blotting.** Samples were homogenized in a Triton X-100 lysis buffer [50 mM Tris-HCl pH = 7.4, 150 mM NaCl, 1% Triton X-100 with 1× protease and phosphatase inhibitors (Pierce)] using sonication and centrifuged at $20,000 \times g$ for 20 min at 4 °C. The supernatant was collected and the total protein concentration was determined using BCA assay (Pierce). 20 μg of proteins were denatured in 1x Laemmli buffer, 2.5% β-mercapto-ethanol for 10 min at 70 °C and loaded in a Criterion™ TGX™ precast midi protein gel (Biorad). Then, a migration at 150 V for 40 min with the manufacturer's buffer (BioRad) was performed. Transfer on LF-PVDF membrane was performed for 7 min at 2.5 A constant, and up to 25 V in the manufacturer buffer using the Trans-blot Turbo machine (BioRad). Membrane was saturated in 5% non-fat dry milk/TBST (20 mM Tris, 150 mM NaCl, 0.1% Tween20, pH = 7.4) for 45 min and then incubated in 5% milk/TBST for 48 h at 4 °C with the following primary antibodies: anti-ACTIN (1/2500, Sigma), anti-TSPO (1/250, Abcam). Following 3 washes in TBST, the membrane was incubated in the appropriated Alexa Fluor-conjugated secondary antibody (1/1000; Invitrogen) in 5% milk/TBST for 90 min. Following 3 washes in TBST, the fluorescence was detected using the iBright imaging system (ThermoFisher Scientific). Densitometry analysis was performed using ImageJ to quantify proteins. Protein levels were normalized to ACTIN levels.

**LC-MSMS analysis of supernatant for XBD173 concentration.** Supernatant samples were stored at −20 °C or lower until analysis. Samples (25 μL) were prepared for analysis by protein precipitation with acetonitrile containing internal standard (tolbutamide) (200 μL) followed by mixing (4 g, 15 min) and centrifugation (1400 × g, 15 min). The supernatant (50) μL was diluted with water (100 μL) and mixed (2 g, 15 min). Samples were analysed by LC-MSMS (Shimadzu Nexera X2 UHPLC/Shimadzu LCMS 8060) with Phenomenex Kinetex Biphenyl (50 × 2.1)mm, 1.7 μm column and mobile phase components water/ 0.1% formic acid (A) and acetonitrile/0.1% formic acid (B). Mobile phase gradient was 0 to 0.3 min 2% B; 0.3 to 1.1 min increase to 95% B; 1.1 to 1.75 min 95% B, 1.75 to 1.8 min decrease to 2% B; 1.8 to 2.5 min 2% B. Flow rate was 0.4 mL/min. Injection volume was 1 μL. Calibration standards were prepared by spiking XDB173 into control supernatant over the range of 2–10,000 ng/mL, then preparing and analysing as for the study samples. Lower limit of detection was 2 ng/mL.

**Statistics and reproducibility.** Immunohistochemical and Imaging Mass Cytometry experiments have been performed with the appropriate technical positive and negative controls. For representative micrographs experiments were repeated a minimum of three times in independent samples. Sample sizes for all experiments were based on availability of human and animal tissues and in line with our previous work using cell cultures and postmortem analysis of human and animal tissues. l statistics have been performed using Graphpad Prism 9.1 and R (v4). Statistical tests have been described per method section as well as detailed in Supplementary Data File 3. Data was considered significant when $P < 0.05$.

### Reporting summary

Further information on research design is available in the Nature Portfolio Reporting Summary linked to this article.

## Data availability

The data that support the findings of this study are available in this manuscript and the Supplementary Information. RNA sequencing data generated in this study can be found https://www.ncbi.nlm.nih.gov/geo/query/acc.cgi?acc=GSE236998, https://www.ncbi.nlm.nih.gov/geo/query/acc.cgi?acc=GSE236999 and https://doi.org/10.26037/yareta:mmopl2vgzrha5ougzm4xftr4ee. All publicly available datasets used in this study have been summarised in Supplementary Table S12[29,30,39–41,50–53,55,57,107–125]. Source data are provided with this paper.

## Code availability

Analysis scripts used in this manuscript are available on GitHub (https://github.com/nfancy/TSPO_Nature_Comm)[126].

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

## Acknowledgements

The authors thank the UK MS Society for financial support (grant number: C008-16.1). DRO was funded by an MRC Clinician Scientist Award (MR/N008219/1). P.M.M. acknowledges generous support from Edmond J Safra Foundation and Lily Safra, the NIHR Senior Investigator programme and the UK Dementia Research Institute which receives its funding from DRI Ltd., funded by the UK Medical Research Council, Alzheimer's Society, and Alzheimer's Research UK. P.M.M. and D.R.O. thank the Imperial College Healthcare Trust-NIHR Biomedical Research Centre for infrastructure support and the Medical Research Council for support of TSPO studies (MR/N016343/1). E.A. was supported by the ALS Stichting (grant "The Dutch ALS Tissue Bank"). Dr. Sally Cowley (Oxford Parkinson's Disease Centre, James Martin Stem Cell Facility, University of Oxford) provided the iPS cell line and expertise in differentiation to iPS-microglia. All authors thank the NIHR Imperial Clinical Research Facility (ICRF) for supporting procedures relating to collection of blood samples. P.M. and B.B.T. are funded by the Swiss National Science Foundation (projects 320030_184713 and 310030_212322, respectively). S.T. was supported by an "Early Postdoc.Mobility" scholarship (P2GEP3_191446) from the Swiss National Science Foundation, a "Clinical Medicine Plus" scholarship from the Prof Dr. Max Cloëtta Foundation (Zurich, Switzerland), from the Jean et Madeleine Vachoux Foundation (Geneva, Switzerland) and from the University Hospitals of Geneva. The authors wish to thank Pia Lovero and Adrien Fischer for expert technical assistance. The results published here are in part based on data obtained from the AD Knowledge Portal (https://adknowledgeportal.org). Study data were generated from postmortem brain tissue provided by the Religious Orders Study and Rush Memory and Aging Project (ROSMAP) cohort at Rush Alzheimer's Disease Center, Rush University Medical Center, Chicago. This work was funded by NIH grants U01AG061356 (De Jager/Bennett), RF1AG057473 (De Jager/Bennett), and U01AG046152 (De Jager/Bennett) as part of the AMP-AD consortium, as well as NIH grants R01AG066831 (Menon) and U01AG072572 (De Jager/St George-Hyslop).

## Author contributions

Conceptualisation: E.N., N.F., M.W., S.T., P.M.M., C.S.M., P.M., S.A., and D.R.O. Technical and analysis support: J.A., M.B., D.S., S.C., M.C.T., T.Saito., T.Saido., O.H.W., M.W., C.S.M., C.B. and C.I.R. Data collection and curation: E.N., N.F., M.W., M.C.M., S.T., R.C.J.M., I.F., J.B., D.H.,

M.F.D., D.A.G., R.Y.B., K.C., A.B., E.K., J.H.W., A.M.Barron, A.M.Badina, A.M.S., B.B.T., J.d.B., R.P., J.A., J.T., K.H.W., P.B-L., A.P., L.H., Q.A., E.A., D.W.H., P.V., S.J., D.B., M.K. and H.K. Writing—original draft: E.N., N.F., S.A. and D.R.O. Writing—review and editing: All authors have reviewed the manuscript. Visualisation: E.N., N.F., M.W., M.C.M., S.T., R.C.J.M., and I.F. Supervision: P.M.M., C.S.M., S.A. and D.R.O.

## Competing interests

The authors declare no competing interests.

## Additional information

[1]Department of Pathology, Amsterdam UMC – Location VUmc, Amsterdam, The Netherlands. [2]Department of Neurobiology and Aging, Biomedical Primate Research Centre, Rijswijk, The Netherlands. [3]Department of Brain Sciences, Imperial College London, London, UK. [4]UK Dementia Research Institute at Imperial College London, London, UK. [5]Department of Psychiatry, University of Geneva, Geneva, Switzerland. [6]Viral Immunology Section, NIH, Bethesda, MD, USA. [7]Flow and Imaging Cytometry Core Facility, NIH, Bethesda, MD, USA. [8]Department of (Neuro)Pathology, Amsterdam UMC, University of Amsterdam, Amsterdam Neuroscience, Amsterdam, The Netherlands. [9]UK Dementia Research Institute at Edinburgh, Edinburgh, UK. [10]Department of Neuroscience, Mario Negri Institute for Pharmacological Research IRCCS, Milan, Italy. [11]Laboratory for Proteolytic Neuroscience, RIKEN Brain Science Institute, Wako-shi, Saitama, Japan. [12]Department of Neurocognitive Science, Institute of Brain Science, Nagoya City University, Nagoya, Japan. [13]Department of Basic and Clinical Neuroscience, Institute of Psychiatry, Psychology and Neuroscience, King's College London, London, UK. [14]Montreal Neurological Institute, McGill University, Montreal, Canada. [15]Division of Biomedical Sciences, Memorial University of Newfoundland, St. John's, Canada. [16]Neurobiology of Aging and Disease Laboratory, Lee Kong Chian School of Medicine, Nanyang Technological University Singapore, Singapore, Singapore. [17]Centre for Brain Research and Department of Pharmacology and Clinical Pharmacology, University of Auckland, Auckland, New Zealand. [18]Institute of Life Science (ILS), Swansea University Medical School, Swansea, UK. [19]Department of Neuroscience and Trauma, Blizard Institute, Queen Mary University of London, London, UK. [20]Institute of Anatomy, Rostock University Medical Center, 18057 Rostock, Germany. [21]Division of Adult Psychiatry, University Hospitals of Geneva, Geneva, Switzerland. [22]These authors contributed equally: Erik Nutma, Nurun Fancy, Maria Weinert, Stergios Tsartsalis. [23]These authors jointly supervised this work: Paul M. Matthews, Craig S. Moore, Sandra Amor, David R. Owen. ✉e-mail: s.amor@amsterdamumc.nl; d.owen@imperial.ac.uk

