## [Peer Review File · Nature Communications]

Translocator protein is a marker of activated microglia in rodent models but not human neurodegenerative diseasesEditorial Note: This manuscript has been previously reviewed at another journal that is not operating a transparent peer review scheme. This document only contains reviewer comments and rebuttal letters for versions considered at Nature Communications.

REVIEWERS' COMMENTS

Reviewer #1 (Remarks to the Author):

The authors have addressed most of my concerns. The study is significantly improved. Considering TSPO PET imaging is commonly used, the study will have a tremendous impact on clinical research of microglia/neuroinflammation in brain diseases. Here are some additional suggestions based on the rebuttal letter:

1. The study clearly showed that TSPO can still label microglia although it is not increased in activated microglia in human tissue. As I commented before, microglial proliferation happens in most brain diseases, and thus TSPO increase may still be relevant to disease states. The authors should tone down their claim.
2. The figures on TSPO western blot as well as vascular TSPO expression should be included in the manuscript (supplementary figures should be fine).
3. The authors should put the figure number in all figures. It is difficult to try to match the figure they mentioned in the rebuttal letter and the manuscript.

Reviewer #2 (Remarks to the Author):

While the authors appropriately responded to the issues raised by this reviewer in their rebuttal letter these are not addressed in the manuscript.

- 1) Fig. S19 is missing
- 2) The reference list is not updated
- 3) The sections in the text they suggested are not incorporated in the manuscript.

It appears that they uploaded a preliminary version of the revision which does not address the comments raised. This has to be corrected.

Reviewer #3 (Remarks to the Author):

Nutma et al. have addressed all of my concerns or suggestions in the last round of review. The authors included additional analyses and text to the main manuscript and supplemental information. To recap, in every round of review, the authors diligently addressed my critiques. I am satisfied with the manuscript; it is much improved and I have no remaining concerns. This manuscript will have a significant impact on the field, and will be of great interest to academia and industry.

I have been asked by the editor to comment on whether the authors have addressed Reviewer #3's concerns. In brief, I am satisfied that the authors sufficiently addressed all of Reviewer #3's concerns—most of which were minor and technical and did not challenge the main findings or conclusions. The authors provided sufficient rebuttal to alleviate the concern and added text to the manuscript or data to supplement to provide clarity. Below, I have featured the major themes of critique by Reviewer #3 and will provide greater detail on the rebuttal.

1. Re: experiments on primary human cells and the source of these cells.

In response to earlier concerns, the authors obtained primary human cells to perform LPS stimulation of cultured human microglia in vitro. The major outcome of this work yielded results consistent with the data from all the fixed tissue staining and quantification. Reviewer #3 criticized that the authors did not include a low dose of LPS. The authors stated that there was a limited source of resected living brain tissue and thus isolated microglia, and so not all conditions could be mirrored. Thus, they chose the high dose and found no TSPO upregulation. I believe this result is sufficient, and we would not learn anything from the low dose experiment.

Further, Reviewer #3 challenged the relevance of using resected brain tissue from epilepsy patients. The authors acknowledged the potential caveat, but the fact of the matter remains that

most biomedical researchers who study primary microglia use such tissues as this is the most “regular” source of living human brain tissue for isolation of microglia. Having said that, it is not trivial to obtain it because it is still limited and in high demand for many types of research. Lastly, even if the authors obtained tissues from the perfect source, the moment cells are in culture their phenotypes change dramatically within 6 hours of plating as seen by recent work from the Glass Lab at UCSD—with this in mind, the concern is moot.

2. Re: the preservation methods, brain region selection and AD stage of the human brain samples selected for this study.

Reviewer 3 had several concerns which were all addressed by the author’s rebuttal. This was largely a clarification of the details of the procedures, which were added to the manuscript text. Also, some of the concerns on AD stage were shared by my previous review, and the authors addressed it comprehensively. Further, the authors examine different brain regions and still arrive at the same conclusion. This additional work and data added to the manuscript satisfy both my and Reviewer #3’s concerns.

3. Re: the use of 125-I autoradiography for TSPO binding in AD tissues.

These experiments are adequate and fulfill the concern originally posed by Reviewer #3. The issue now is that Reviewer #3 would have expected to see these experiments done with tritium labeled compound, not 125-I. Reviewer #3 did not specify in the last round of review. The authors researched the methodology and cite literature showing that 125-I and 3-H labeled TSPO probes show comparable binding and autoradiography. Both techniques are used as standard application in the field. The tritium autoradiography may afford better spatial resolution, but is still far from cellular resolution, so there is really no point. The 125-I method still shows regional analysis which the authors quantify appropriately using standard practice in the field. This reviewer has no concerns of the data presented in this section.

4. Re: several concerns about the procedural details of image analysis or single cell/nuclei analysis.

All of Reviewer #3’s concerns were addressed by the author’s rebuttal and additional text were added to the methods and figure legends to clarify the details.

5. Re: The inclusion of the LCP2 and TFEC findings as novel markers of human microglial specific activation.

Reviewer #3 argues that this data should be excluded because it is not relevant to the current study. However, I disagree—and agree with the authors. Reviewer #2 and I both find this data to be significant and relevant to this manuscript.

REVIEWERS' COMMENTS

Reviewer #1 (Remarks to the Author):

The authors have addressed most of my concerns. The study is significantly improved. Considering TSPO PET imaging is commonly used, the study will have a tremendous impact on clinical research of microglia/neuroinflammation in brain diseases. Here are some additional suggestions based on the rebuttal letter:

1. The study clearly showed that TSPO can still label microglia although it is not increased in activated microglia in human tissue. As I commented before, microglial proliferation happens in most brain diseases, and thus TSPO increase may still be relevant to disease states. The authors should tone down their claim. **We were careful to say in the initial manuscript that our message was not that TSPO PET lacks utility, but rather our interpretation of the signal was imperfect. We re-iterated that point in the manuscript in response to this reviewer's comment. We are therefore not sure what the reviewer means by "tone down their claim" and so we do not propose to edit the manuscript further.**
2. The figures on TSPO western blot as well as vascular TSPO expression should be included in the manuscript (supplementary figures should be fine). **We have now included these results (Figure S7 and statistics table)**
3. The authors should put the figure number in all figures. It is difficult to try to match the figure they mentioned in the rebuttal letter and the manuscript. **We apologize for this, figures will of course be numbered at the published version.**

Reviewer #2 (Remarks to the Author):

While the authors appropriately responded to the issues raised by this reviewer in their rebuttal letter these are not addressed in the manuscript.

- 1) Fig. S19 is missing. **We included it in the reviewer comments as we were not sure whether reviewer 1 would want it in the manuscript. We have now included it in the manuscript (Fig S21).**
- 2) The reference list is not updated **Apologies. As we said at the time, this was because the author who held the references was on leave. We did not feel it was sensible to delay resubmission for that reason.**
- 3) The sections in the text they suggested are not incorporated in the manuscript. **Again, this is because these sections were suggested edits. We were not sure whether the reviewers thought we should include them in the text itself. We have now added them**

It appears that they uploaded a preliminary version of the revision which does not address the comments raised. This has to be corrected. **Again, as above, the manuscript we sent was the “final” revised manuscript, but there were a few instances where we included suggested edit and figures in the reviewer responses but not the manuscript. This was because we did not know at the time whether the reviewers would think they were necessary. Now that the reviewers have made clear they do want them, we have now added them all to the manuscript.**

Reviewer #3 (Remarks to the Author):

This reviewer was asked to comment not only on our response to his/her review, but also on our response to reviewer #3. This reviewer was satisfied with our responses and has not asked for anything else.

Nutma et al. have addressed all of my concerns or suggestions in the last round of review. The authors included additional analyses and text to the main manuscript and supplemental information. To recap, in every round of review, the authors diligently addressed my critiques. I am satisfied with the manuscript; it is much improved and I have no remaining concerns. This manuscript will have a significant impact on the field, and will be of great interest to academia and industry.

I have been asked by the editor to comment on whether the authors have addressed Reviewer #3's concerns. In brief, I am satisfied that the authors sufficiently addressed all of Reviewer #3's concerns—most of which were minor and technical and did not challenge the main findings or conclusions. The authors provided sufficient rebuttal to alleviate the concern and added text to the manuscript or data to supplement to provide clarity. Below, I have featured the major themes of critique by Reviewer #3 and will provide greater detail on the rebuttal.

1. Re: experiments on primary human cells and the source of these cells.

In response to earlier concerns, the authors obtained primary human cells to perform LPS stimulation of cultured human microglia in vitro. The major outcome of this work yielded results consistent with the data from all the fixed tissue staining and quantification. Reviewer #3 criticized that the authors did not include a low dose of LPS. The authors stated that there was a limited source of resected living brain tissue and thus isolated microglia, and so not all conditions could be mirrored. Thus, they chose the high dose and found no TSPO upregulation. I believe this result is sufficient, and we would not learn anything from the low dose experiment. Further, Reviewer #3 challenged the relevance of using resected brain tissue from epilepsy patients. The authors acknowledged the potential caveat, but the fact of the matter remains that most biomedical researchers who study primary microglia use such tissues as this is the most “regular” source of living human brain tissue for isolation of microglia. Having said that, it is not trivial to obtain it because it is still limited and in high demand for many types of research. Lastly, even if the

authors obtained tissues from the perfect source, the moment cells are in culture their phenotypes change dramatically within 6 hours of plating as seen by recent work from the Glass Lab at UCSD— with this in mind, the concern is moot.

2. Re: the preservation methods, brain region selection and AD stage of the human brain samples selected for this study.

Reviewer 3 had several concerns which were all addressed by the author's rebuttal. This was largely a clarification of the details of the procedures, which were added to the manuscript text. Also, some of the concerns on AD stage were shared by my previous review, and the authors addressed it comprehensively. Further, the authors examine different brain regions and still arrive at the same conclusion. This additional work and data added to the manuscript satisfy both my and Reviewer #3's concerns.

3. Re: the use of 125-I autoradiography for TSPO binding in AD tissues.

These experiments are adequate and fulfill the concern originally posed by Reviewer #3. The issue now is that Reviewer #3 would have expected to see these experiments done with tritium labeled compound, not 125-I. Reviewer #3 did not specify in the last round of review. The authors researched the methodology and cite literature showing that 125-I and 3-H labeled TSPO probes show comparable binding and autoradiography. Both techniques are used as standard application in the field. The tritium autoradiography may afford better spatial resolution, but is still far from cellular resolution, so there is really no point. The 125-I method still shows regional analysis which the authors quantify appropriately using standard practice in the field. This reviewer has no concerns of the data presented in this section.

4. Re: several concerns about the procedural details of image analysis or single cell/nuclei analysis.

All of Reviewer #3's concerns were addressed by the author's rebuttal and additional text were added to the methods and figure legends to clarify the details.

5. Re: The inclusion of the LCP2 and TFEC findings as novel markers of human microglial specific activation.

Reviewer #3 argues that this data should be excluded because it is not relevant to the current study. However, I disagree—and agree with the authors. Reviewer #2 and I both find this data to be significant and relevant to this manuscript.